# GPR182 limits antitumor immunity via chemokine scavenging in mouse melanoma models

Robert J. Torphy [1], Yi Sun[1], Ronggui Lin[1], Alayna Caffrey-Carr[2], Yuki Fujiwara [1], Felix Ho[1], Emily N. Miller [1], Martin D. McCarter[1], Traci R. Lyons [3], Richard D. Schulick[1], Ross M. Kedl[2] & Yuwen Zhu [1✉]

For many solid tumors, immune checkpoint blockade therapy has become first line treatment, yet a large proportion of patients with immunologically cold tumors do not benefit due to the paucity of tumor infiltrating lymphocytes. Here we show that the orphan G Protein-Coupled Receptor 182 (GPR182) contributes to immunotherapy resistance in cancer via scavenging chemokines that are important for lymphocyte recruitment to tumors. GPR182 is primarily upregulated in melanoma-associated lymphatic endothelial cells (LECs) during tumorigenesis, and this atypical chemokine receptor endocytoses chemokines promiscuously. In GPR182-deficient mice, T cell infiltration into transplanted melanomas increases, leading to enhanced effector T cell function and improved antitumor immunity. Ablation of GPR182 leads to increased intratumoral concentrations of multiple chemokines and thereby sensitizes poorly immunogenic tumors to immune checkpoint blockade and adoptive cellular therapies. CXCR3 blockade reverses the improved antitumor immunity and T cell infiltration characteristic of GPR182-deficient mice. Our study thus identifies GPR182 as an upstream regulator of the CXCL9/CXCL10/CXCR3 axis that limits antitumor immunity and as a potential therapeutic target in immunologically cold tumors.

[1] Department of Surgery, University of Colorado Anschutz Medical Campus, Aurora, CO 80045, USA. [2] Department of Immunology and Microbiology, University of Colorado Anschutz Medical Campus, Aurora, CO 80045, USA. [3] Division of Medical Oncology, Department of Medicine, University of Colorado Anschutz Medical Campus, Aurora, CO 80045, USA. ✉email: yuwen.zhu@cuanschutz.edu

Immune checkpoint blockade (ICB) therapy has dramatically improved survival for many patients with advanced-stage melanoma; however, objective response rates remain <40% in patients with advanced disease[1–3]. One of the main reasons for resistance to ICB therapy is that many melanoma tumors are immunologically cold and lack effector T cell infiltration into the tumor microenvironment (TME)[4]. Consistent with this, CD8+ effector T cell density within melanoma predicts responsiveness to anti-PD1 therapy[5]. New therapeutic targets are greatly needed to improve CD8+ T cell infiltration into the TME in order to improve response rates to ICB therapy.

Chemokines, and their receptors, play a crucial role in anti-tumor immunity as they regulate immune cell homing into tumors[6]. By attaching to glycosaminoglycans (GAGs) on endo-thelial cells (ECs) and in the extracellular matrix (ECM), che-mokines produced at the tumor site create gradients that guide T cells into the tumor[7]. Among them, CXCR3 and its ligands, CXCL9, CXCL10, and CXCL11, play a central role in regulating effector T cell infiltration into tumors and are essential for effective ICB therapy[8–10]. Effector CD8+ T cells upregulate CXCR3 and migrate into the TME in response to gradients of its cognate ligands, which are mainly produced by CD103+ den-dritic cells (DC) and macrophages within tumors. CXCL9-11 expression is further induced in response to IFN-γ generated by intratumoral CD8+ T cells[9,10]. In human cancers, the expression levels of CXCR3 ligands are positively associated with the num-bers of infiltrating CD8+ T cells and improved survival of cancer patients[11,12].

Atypical chemokine receptors (ACKRs) are a group of GPCRs that function as scavenger receptors to restrain tissue chemokine levels[13,14]. Unlike chemokine receptors present on leukocytes, all known ACKRs are found to be expressed by ECs and other stromal cell types. ACKRs have mutations at the DRYLAIV motif that is essential for G protein coupling and classical GPCR signaling[15,16]. ACKRs function to scavenge multiple chemokine ligands, shape chemokine gradients in the extracellular space, and signal through β-arrestin[15,16]. A recent publication found that loss of ACKR4 increases intratumoral CCL21 to promote the retention of migratory CD103+ DCs, which resulted in improved antitumor immunity[17], supporting that ACKRs can be feasible therapeutic targets to improve responsiveness to immunotherapy.

G Protein-Coupled Receptor 182 (GPR182) (also named as ADMR) is an orphan Class A GPCR with homology to the chemokine receptor family[18]. GPR182 is found to be pre-ferentially expressed in vascular ECs, liver sinusoidal ECs, LECs, as well as intestinal stem cells[19–22]. GPR182 transcript is reported to be upregulated in tumor-associated ECs[23,24], however, the function of GPR182 remains poorly described.

In this study, we investigate the role of GPR182 in the anti-tumor immune response. Our studies in mouse melanoma models support that GPR182 serves as an ACKR in the TME to scavenge chemokines to limit T cell infiltration.

## Results

### GPR182 is upregulated on lymphatic endothelial cells (LECs) within human melanoma.
To determine GPR182 expression in melanoma, we queried single-cell expression data from two recent studies in human melanoma[25,26]. We found that GPR182 expression is restricted to ECs in both datasets (Fig. 1a). Detailed analyses of the EC population revealed that GPR182 is exclusively expressed in lymphatic ECs (LECs) (Fig. 1b). We further char-acterized GPR182 protein expression by immunofluorescent staining of human melanoma. We observed that only a small subset of weakly CD31+ vessels in the TME were positive for GPR182 (Fig. 1c, upper panel). Costaining of serial sections for

GPR182 and podoplanin (PDPN), a specific marker for LECs, identified that GPR182 was exclusively expressed on PDPN+ lymphatic vessels in human melanoma (Fig. 1c, middle panel). Consistently, GPR182 protein was co-expressed with Prox-1, another protein primarily expressed in human LECs (Fig. 1c, lower panel)[27]. In human breast cancer, we also found that GPR182 expression was restricted to PDPN+ LECs (Supple-mentary Fig. 1a). To query the association of lymphatic endo-thelium and the expression of GPR182 across a larger sample of human melanoma tumors, we queried 288 metastatic melanoma samples from The Cancer Genome Atlas (TCGA). We generated a lymphatic score for melanoma samples based on the relative expression levels of PDPN, LYVE1, and VEGFC[28], and found that GPR182 expression significantly correlated with lymphatic score in melanoma (Fig. 1d).

To compare the expression of GPR182 in melanoma-associated LECs versus patient-matched normal skin, we performed multi-plex staining in primary tumor tissues from patients with stage II/III melanoma. We observed strong GPR182 staining in PDPN-positive lymphatic vessels within the tumor, while the expression of GPR182 was weak or negative in lymphatics of the adjacent normal skin (Fig. 1e). In normal human skin, GPR182 was undetectable in LECs (Supplementary Fig. 1b). To quantify the expression of GPR182 on lymphatics we measured mean fluorescent intensity (MFI) of GPR182 on PDPN+ lymphatic vessels; we observed a marked increase in GPR182 expression in melanoma-associated lymphatic vessels compared to lymphatics in adjacent normal skin (Fig. 1f).

### Loss of GPR182 triggers improved antitumor immunity to slow tumor progression.
To investigate if GPR182 affects tumor growth, we obtained the $Gpr182^{tm2q(KOMP)Wtsi/+}$ ($GPR182^{lacZ/+}$) mouse (hereafter referred to as the GPR182+/− mouse) from the Knockout Mouse Project (KOMP) repository and backcrossed to achieve homozygous $GPR182^{lacZ/lacZ}$ (hereafter referred to as GPR182−/−) mice as described by Kechele et al.[22]. First, we per-formed immunofluorescent staining to examine the expression of mouse GPR182 in B16, YUMM1.7, and YUMMER1.7 tumors, three mouse melanoma models. Similar to human GPR182, the expression of mouse GPR182 on LYVE-1-positive LECs in normal skin is limited; however, it is strongly expressed in tumor-associated LECs, and LECs seemed to be the main source for GPR182 in tumors (Supplementary Fig. 1c). We inoculated GPR182−/−, GPR182+/−, and wildtype (WT) littermates subcutaneously with B16 melanoma and monitored tumor progression. GPR182−/− mice displayed significantly slower tumor outgrowth than WT and GPR182+/− littermates (Fig. 2a); as a result, tumor mass in GPR182−/− mice at the study endpoint was reduced threefold compared to WT and twofold compared to GPR182+/− mice (Fig. 2b). GPR182−/− mice also displayed slower tumor outgrowth than WT mice following subcutaneous inoculation with YUMM1.7 murine melanoma (Fig. 2c, d). As B16 and YUMM1.7 tumors are both poorly immunogenic, we challenged mice with a more immunogenic YUMMER1.7 cell line[29]. The growth of YUM-MER1.7 was markedly reduced in GPR182−/− mice compared to WT controls, resulting in a fourfold reduction in tumor volume in the GPR182−/− group 14 days after tumor inoculation (Fig. 2e). As a result, GPR182−/− mice displayed significantly longer overall survival than WT controls, with 3 of 11 GPR182−/− mice achieving complete tumor regression (Fig. 2f). GPR182−/− mice that exhibited complete tumor regression gained immunological memory as they were resistant to YUMMER1.7 re-challenge with a larger tumor inoculum (Fig. 2g).

To investigate whether the reduced tumor growth in GPR182−/− mice was due to improved antitumor T cell response, we depleted

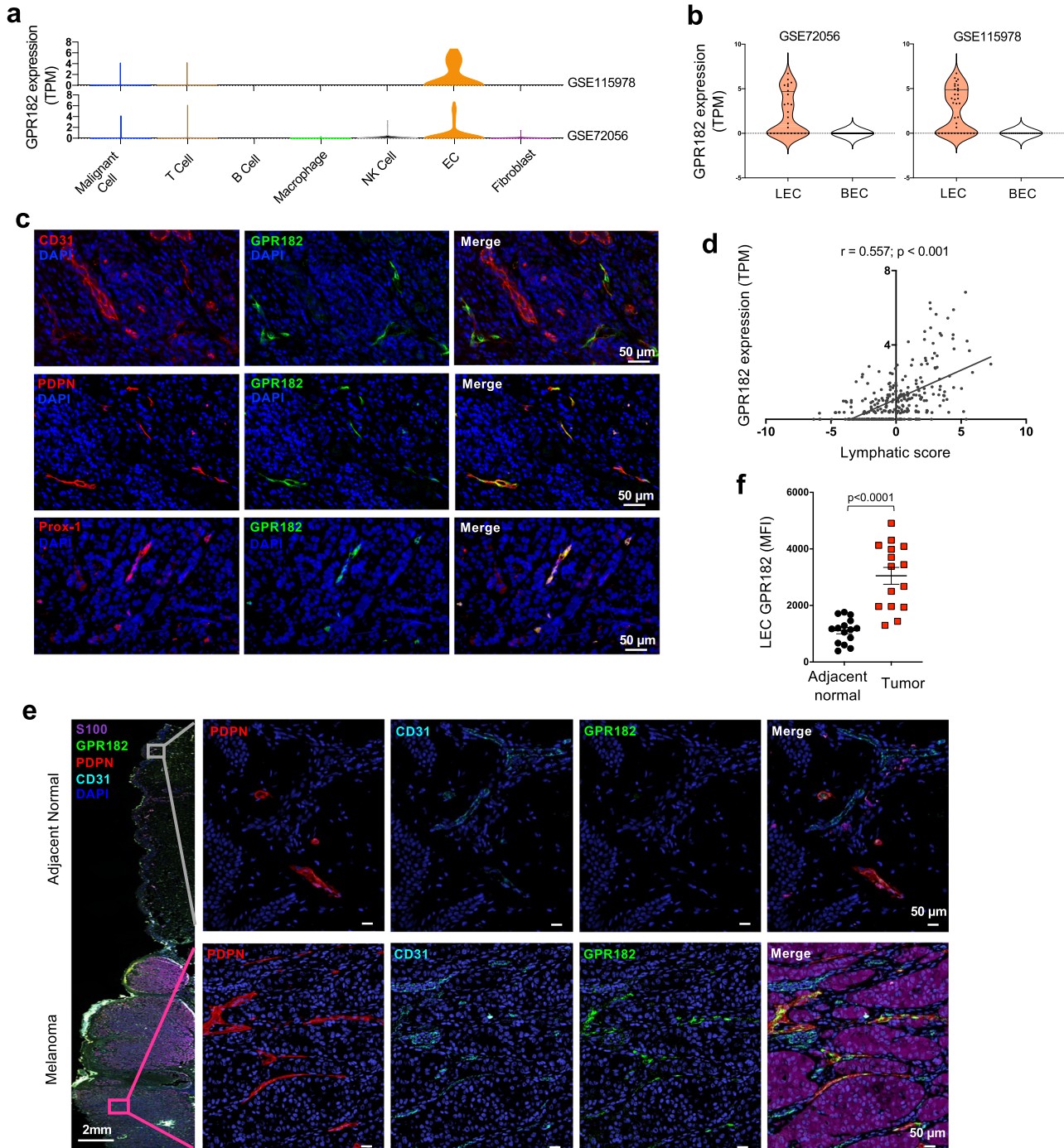

**Fig. 1 GPR182 is expressed by lymphatic endothelial cells in human melanoma. a, b** Two published single-cell RNA sequencing datasets of human melanoma were queried for GPR182 expression based on cell type (**a**). **b** Analysis of ECs in human melanoma further revealed that *GPR182* is primarily expressed in LECs. **c** Human melanoma tissues were stained for GPR182 (green) together with EC markers, including CD31, podoplanin (PDPN), or Prox-1. **d** A lymphatic score was generated using 288 metastatic melanoma samples from the TCGA database. The lymphatic score was calculated based on relative expression levels of *PDPN*, *LYVE1*, and *VEGFC* in each sample. Lymphatic score was plotted against GPR182 mRNA expression level (RSEM, log2 normalized). Pearson's correlation coefficient, *r*, and *p*-values shown from two-sided test. **e, f** Human melanoma tissues with adjacent normal skin were stained for PDPN (red), CD31(blue), and GPR182 (green) (E); S-100 staining (brown) was used to identify melanocytes and tumor cells. **f** Quantification of GPR182 median fluorescent intensity (MFI) in PDPN+ lymphatic vessels from paired tumor and adjacent normal tissue. *n* = 15, *P*-value from two-sided paired *t*-test. Error bars represent SEM.

CD4+ and CD8+ T cells in GPR182−/− and WT mice by intraperitoneal injection of anti-CD4 and anti-CD8β depleting mAbs together. The depletion of T cells was confirmed by flow cytometry staining of peripheral blood immune cells (Supplementary Fig. 2a). As shown in Fig. 2h and i, the removal of T cells in the

B16 and YUMM1.7 tumor models reversed the tumor growth suppression observed in GPR182−/− mice.

To comprehensively characterize intratumoral immune cells, we dissected YUMM1.7 tumor tissues and performed flow cytometry analysis (Supplementary Fig. 2b, c). YUMM1.7 tumors

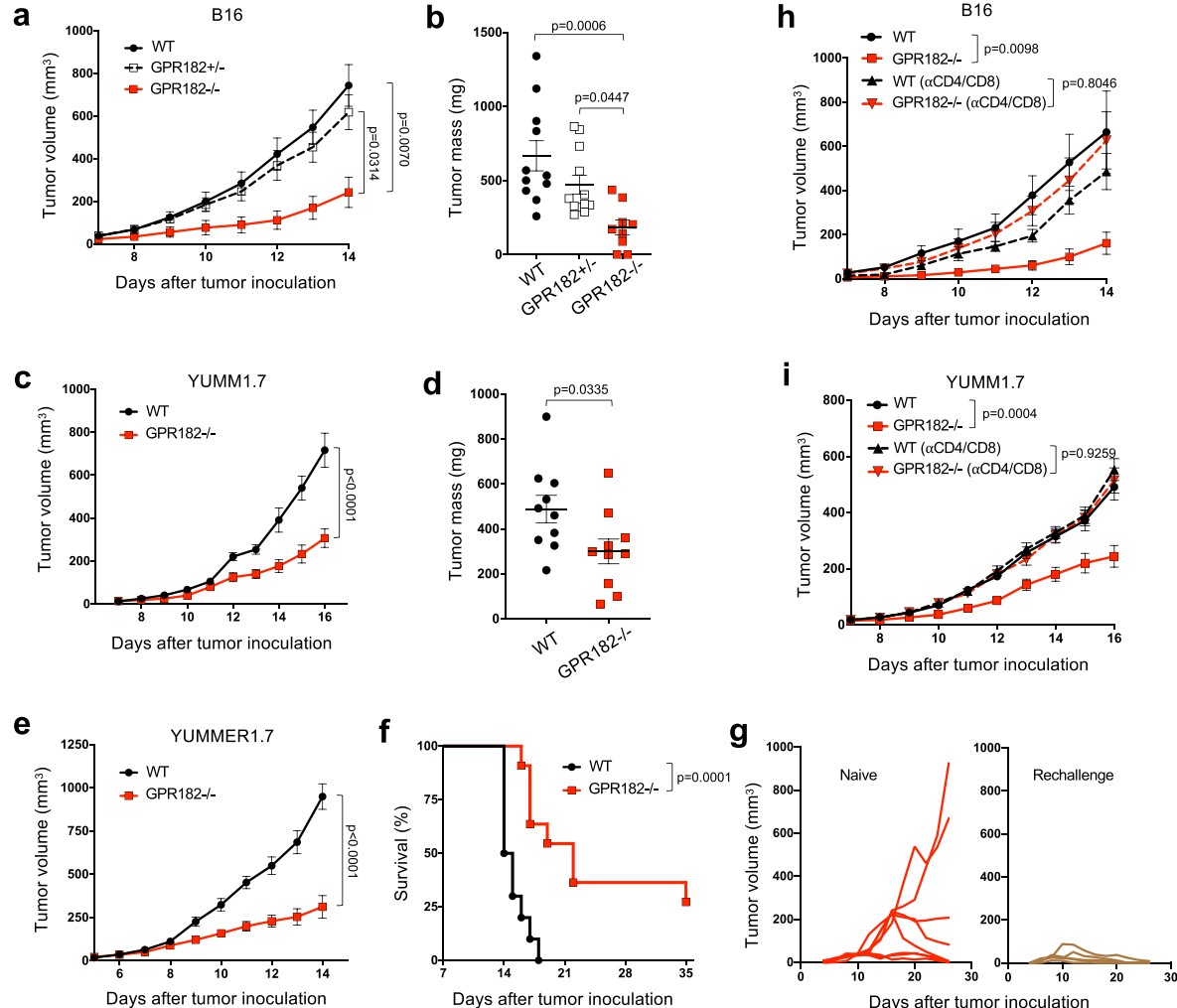

**Fig. 2 GPR182−/− mice exhibit improved antitumor immunity. a**, **b** GPR182−/− and littermates were inoculated subcutaneously with 200,000 B16 tumor cells. Tumor growth curves (**a**) and tumor mass 14 days after tumor inoculation (**b**) were recorded. n = 11, 12, and 9, respectively, per group. Representative data from two independent experiments. **c**, **d** WT and GPR182−/− mice were inoculated subcutaneously with 50,000 YUMM1.7 tumor cells. Tumor growth curves (**c**) and tumor mass 16 days after tumor inoculation (**d**) were recorded. n = 9–10 per group. Representative data from three independent experiments. **e**–**g** Tumor growth curves (**e**) and Kaplan–Meier curves (**f**) for the survival of WT and GPR182−/− mice inoculated subcutaneously with 250,000 YUMMER1.7 tumor cells. n = 10–11 per group. Representative data from two independent experiments. Tumor-free GPR182−/− mice (n = 5) were challenged with YUMMER1.7 one month after tumors were completely rejected; naive GPR182−/− mice (n = 8) were used as comparison (**g**). **h** B16 tumor growth curves in WT and GPR182−/− mice upon T cell depletion. In the T cell depletion groups, mice were treated with αCD4 and αCD8 depleting antibodies 1 day prior to tumor inoculation and again at day 7 after tumor inoculation. n = 10 per group. **i** YUMM1.7 tumor growth curves in WT and GPR82−/− mice upon T cell depletion. In the T cell depletion groups, mice were treated with αCD4 and αCD8 depleting antibodies 1 day prior to tumor inoculation and again at day 7 after tumor inoculation. n = 10 per group. P-values for growth curves from two-way ANOVA test (**a**, **c**, **e**, **h**, **i**). P-values for tumor mass from one-way ANOVA with multiple comparisons (**b**) and two-sided student's t-test (**d**). P-values for Kaplan–Meier curves from log-rank test (**f**). Error bars represent SEM.

from GPR182−/− mice demonstrated significantly increased CD45+ immune cells and CD3+ T cells both in proportions and in densities (Fig. 3a, b, and Supplementary Fig. 3a); except for the small group of polymorphonuclear myeloid-derived suppressor cells (PMN-MDSC) (Supplementary Fig. 3b), we did not observe any significant difference in non-T cell populations, including natural killer (NK) cells, tumor-associated macrophages (TAMs), DCs or total MDSC (Fig. 3b). Our immunofluorescent staining of YUMM1.7 tumor tissues from WT and GPR182−/− hosts verified increased CD3+ T cell infiltrates in tumors from GPR182−/− mice (Fig. 3c). Characterization of the immune cell infiltrates in B16 tumors recapitulated the findings from the YUMM1.7 model, with selective increases of CD3+ T cells in GPR182−/− tumors (Fig. 3d, Supplementary Fig. 3c).

We further examined infiltrating T cells in YUMM1.7 tumors of GPR182−/− mice. Both CD8+ T cells and Foxp3-negative conventional CD4+ T cells were increased in density in the tumors of GPR182−/− mice (Fig. 3e, f). Tumors of GPR182−/− mice also contained significantly more CD8+ T cells producing effector cytokine IFN-γ (Fig. 3g) and Granzyme B (GzmB) (Fig. 3h), indicating an improved effector T cell function. Consistently, YUMM1.7 tumors from GPR182−/− mice had higher frequencies of effector T cells (CD44 + CD62L−) in both CD4+ and CD8+ T cell groups (Fig. 3i). In addition, there were proportionally more CD8+ T cells expressing both PD1 and 4-1BB in GPR182−/− tumors compared to WT controls (Fig. 3j). FoxP3+ Treg cells in total CD4+ T cells were proportionally reduced in GPR182−/− tumors (Fig. 3k). These data suggest that

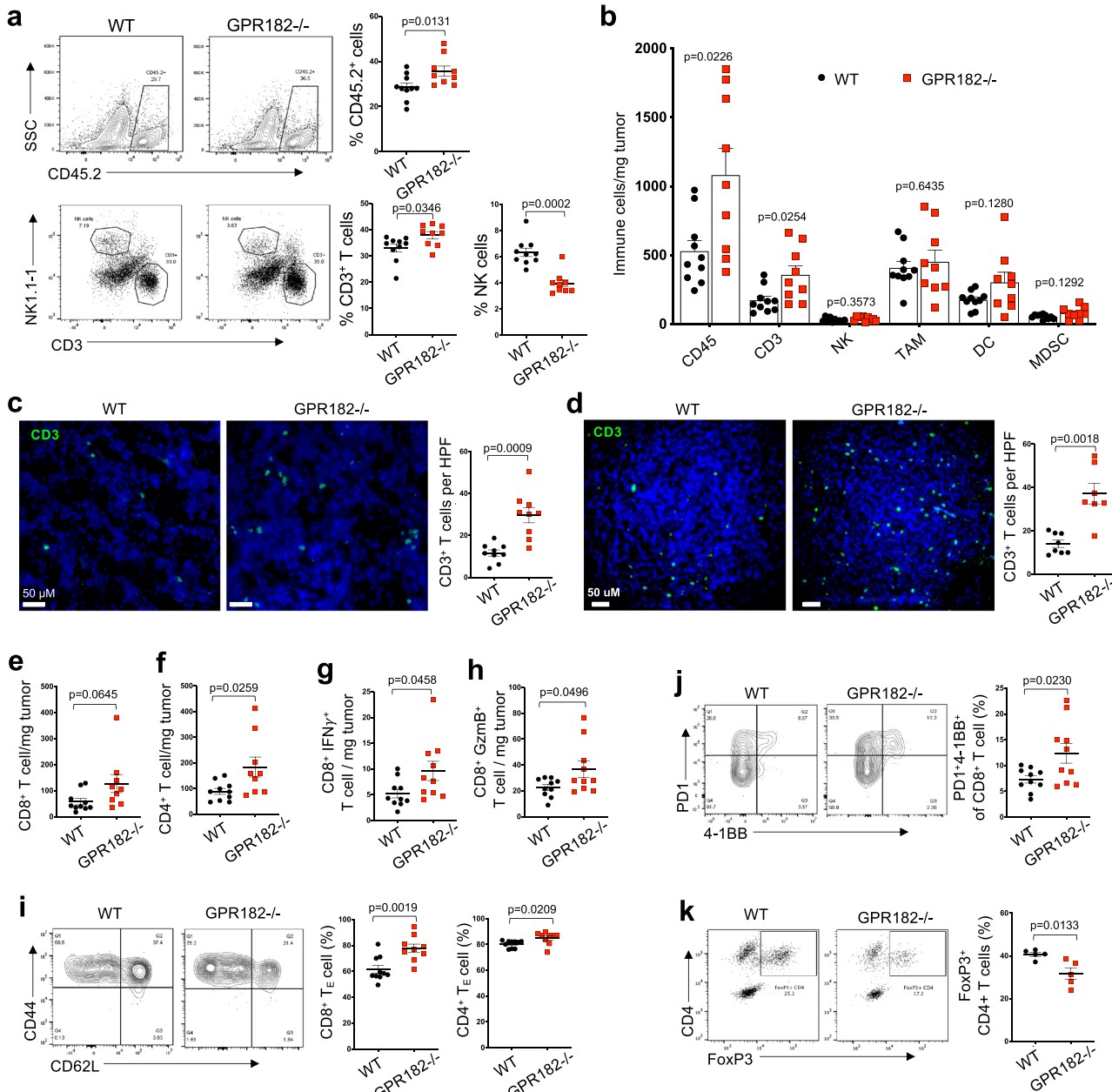

**Fig. 3 Increased T cell infiltration and functions in tumors of GPR182−/− mice. a**, **b** 16 days after tumor inoculation, the immune TME of YUMM1.7 tumors in WT and GPR182−/− mice were characterized. Single-cell suspensions were stained for infiltrating immune cells (CD45+); the percentages of infiltrating T and NK cells were further determined (**a**). The densities of different immune cell types within tumors were quantified by flow cytometry (**b**). $n = 10$ and 9 per group, respectively. Data representative from two independent experiments. **c**, **d** Representative immunofluorescence staining for CD3+ T cells in YUMM1.7 (**c**) and B16 (**d**) tumors from WT and GPR182−/− mice. The densities of CD3+ T cells per high power field (HPF) were quantified. YUMM1.7, $n = 10$ per group. B16, $n = 8$ and 7 per group, respectively. Data representative from two independent experiments. **e**–**k** In YUMM1.7 tumors 16 days after inoculation, the densities of CD8+ (**e**) and CD4+ T cells (**f**), as well as IFNγ- (**g**) or Granzyme B (GzmB) (**h**)- producing CD8+ T cells, were enumerated. The frequencies of effector cells (CD44highCD62L-) in CD8+ and CD4+ T cell population (**i**), and CD8+ T cells expressing both PD1 and 4-1BB (**j**), as well as FoxP3+ Treg cells (**k**), were quantified. **e**–**i** $n = 9$ and 10 per group; **k** $n = 5$ per group. Data representative from two independent experiments. $P$-values from two-sided Student's $t$-test (**a**–**k**). Error bars represent SEM.

the increased number of intratumoral T cells in GPR182−/− host is mainly caused by an increase in effector T cells. In B16 tumors of GPR182−/− mice, there were significant increases of both conventional CD8+ and CD4+ T cells (Supplementary Fig. 3d); T cells with effector phenotype (Supplementary Fig. 3e) and PD1-expressing T cells (Supplementary Fig. 3f) were proportionally increased in both CD8+ and CD4+ T cell subsets within GPR182−/− tumors. In tumor naive mice, the CD3+ T cell densities in normal skin between WT and GPR182−/− mice were

similarly scarce (Supplementary Fig. 3g), suggesting that the increased T cell infiltration in tumors of GPR182−/− mice was not attributed to skin resident T cells.

**Immune phenotype in naïve GPR182−/− mice.** The increased antitumor T cell response observed in tumors from GPR182−/− mice led us to examine T cells in naïve GPR182−/− mice. Similar to previous reports[20,22], we observed increased spleen mass in

naïve adult GPR182−/− mice (Supplementary Fig. 4a, b), presumably due to a negative role of GPR182 in hematopoiesis[30]. We analyzed the cellular composition of spleens by flow cytometry and observed no difference in the percentages of CD3+ T cells, CD19+ B cells, or CD45-negative stromal cells, between WT and GPR182−/− mice. Similarly, we observed no differences in the cellular composition of lymph nodes between the two groups (Supplementary Fig. 4c). We further characterized phenotypes of peripheral blood CD4+ and CD8+ T cells from WT and GPR182−/− mice. There were no differences in the proportions of naïve (CD44$^{low}$ CD62L+), central memory (CD44$^{high}$ CD62L+), and effector memory (CD44$^{high}$ CD62L−) populations between 10-week-old WT and GPR182−/− mice (Supplementary Fig. 4d, 4e). We then isolated T cells from lymph nodes and labeled them with a cell proliferation tracing dye (CFSE) prior to stimulation in vitro with plate-bound anti-CD3 mAb. We did not observe any noticeable difference between WT littermates and GPR182−/− mice regarding the proliferation of either CD4+ (Supplementary Fig. 4f and 4g) or CD8+ T cells (Supplementary Fig. 4h, 4i) across varying concentrations of anti-CD3. Therefore, our study supports that loss of GPR182 does not directly affect T cell functions in tumor naïve mice.

**GPR182−/− mice exhibit increased T cell homing to the TME.** To determine whether reduced tumor growth observed in GPR182−/− mice was mediated by the loss of GPR182 in hematopoietic cells or non-hematopoietic stromal cells, we performed a bone marrow (BM) chimera experiment where GPR182−/− and WT controls were lethally irradiated before reconstitution with BM cells from GPR182−/− or WT donor mice generating four different groups (Fig. 4a). Twelve weeks later when the immune system was fully established, chimeric mice were challenged with B16 melanoma subcutaneously. WT mice reconstituted with GPR182−/− BM displayed similar tumor growth curves to those control mice with WT BM. Tumor growth was similarly restrained in GPR182−/− mice reconstituted with either WT or GPR182−/− BM cells, compared to WT mice reconstituted with WT or GPR182−/− BM cells (Fig. 4b). The two groups of GPR182−/− host mice also displayed increased overall survival compared to WT hosts (Fig. 4c). Therefore, loss of GPR182 in stromal cells, presumably LECs, is likely the primary mechanism driving the attenuated tumor outgrowth in GPR182−/− mice.

The lymphatic system plays a critical role in antigen trafficking, antigen presentation, and the coordination of T cell priming[30,31]. Therefore, we evaluated the possible impact of GPR182 on lymphatic draining function and T cell priming. Measurement of soluble lymph trafficking of FITC-dextran by footpad administration demonstrated no difference in lymphatic drainage between WT and GPR182−/− mice (Supplementary Fig. 4j). To assess T cell priming, B16-OVA tumors were inoculated at a high tumor burden and we observed no difference in tumor outgrowth between WT and GPR182−/− mice. After 10 days of tumor growth, we transferred naïve CFSE- labeled OT-1 T cells (CD45.1+) into B16-OVA tumor-bearing mice to measure tumor-specific T cell priming (Fig. 4d). We observed no difference in the percentages of divided CD45.1+ OT-1 cells in the tumor-draining lymph nodes (dLNs) between WT and GPR182−/− mice (Fig. 4e). In addition, there were no differences in the frequency of CD103 + DCs in tumor dLNs (Fig. 4f) or in the frequency of CD103 + DCs cross-presenting MHC class I (H-2K$^b$) bound OVA$_{257-264}$ (SIINFEKL) in tumor dLNs (Fig. 4g). We prepared single-cell suspensions from dLNs from B16-OVA bearing mice and co-cultured them with naïve CFSE-labeled CD45.1+ OT-1 cells in vitro. We observed no difference in OT-1 cell proliferation

in response to the stimulation of dLN cells between WT and GPR182−/− groups after 72 h of in vitro co-culture (Fig. 4h). Together, these data demonstrate that loss of GPR182 has no impact on antigen trafficking or T cell priming in tumor dLNs.

To evaluate homing of activated T cells in tumors, in vitro activated CD45.1+ OT-1 T cells were intravenously transferred into WT and GPR182−/− mice bearing size-matched B16-OVA tumors. 24 h after intravenous OT-1 transfer, intratumoral OT-1 cells were quantified by flow cytometry. GPR182−/− mice displayed a higher density of intratumoral OT-1 T cells than WT control mice (Fig. 4i, j). To further determine whether the increased density of activated OT-1 cells was solely due to improved homing to tumors in GPR182−/− mice, we tracked T cell division by treating mice with BrdU prior to tissue harvest for flow cytometry analysis (Fig. 4k, l). GPR182−/− mice again displayed a higher density of intratumoral OT-1 cells than WT controls, though the frequency of OT-1 cells in tumor dLNs was similar (Fig. 4m). However, there was no difference between WT and GPR182−/− mice in the percentages of proliferating OT-1 cells in tumors or tumor dLNs, as measured by BrdU incorporation (Fig. 4n). These data demonstrate that increased T cell homing to tumors, not improved T cell priming or proliferation, is responsible for the increased T cell density observed in tumors of GPR182−/− mice.

**Upregulation of the CXCR3 pathway contributes to restrained tumor growth in GPR182−/− mice.** Chemokines are key drivers of immune cell homing through their interaction with chemokine receptors expressed on immune cells[14]. GPR182 is an orphan GPCR receptor closely homologous to ACKR3 (Supplementary Fig. 5a). Similar to other known ACKRs, GPR182 does not have a conserved DRYLAIV motif (Supplementary Fig. 5b) and is expressed primarily by stromal cells, particularly LECs in human melanoma (Fig. 1 and Supplementary Fig. 5c). We hypothesized that GPR182 may act as an ACKR to modulate chemokine bioavailability in the TME[32]. We harvested YUMM1.7 tumors from WT and GPR182−/− mice to measure intratumoral concentrations of 13 chemokines using the LEGENDplex Mouse Chemokine Panel (BioLegend). Multiple chemokines, including CCL2, CCL22, CXCL1, CXCL9, and CXCL10, were significantly increased in tumors of GPR182−/− mice (Fig. 5a).

Chemokines CXCL9 and CXCL10 are well described mediators for activated T cell homing in melanoma through their interaction with receptor CXCR3[33,34]. In our mouse tumor models, functional CXCL11, the third ligand for CXCR3, is lacking in C57BL/6 mice[35]. CXCR3-positive cells in both CD8+ and CD4+ T cell subsets were selectively increased in YUMM1.7 tumors from GPR182−/− mice (Fig. 5b, c). To further evaluate whether the reduced tumor growth and increased T cell density observed in GPR182−/− mice were dependent on the CXCR3 pathway, we treated mice with a CXCR3 blocking mAb throughout YUMM1.7 tumor inoculation. Anti-CXCR3 treatment completely abrogated the attenuated tumor growth observed in GPR182−/− mice, suggesting that the improved anti-tumor immunity seen in GPR182−/− mice is dependent on the CXCL9/CXCL10/CXCR3 axis (Fig. 5d). Analysis of TILs in WT and GPR182−/− mice further demonstrated that anti-CXCR3 treatment markedly reduced T cell infiltration in YUMM1.7 tumors; as a result, the increased intratumoral T cell density observed in tumors from GPR182−/− mice was completely abolished by the treatment of anti-CXCR3 (Fig. 5e and f).

**GPR182 acts as an ACKR for the CXCR3 ligands.** To directly test whether GPR182 acts as an ACKR, we first analyzed the cell

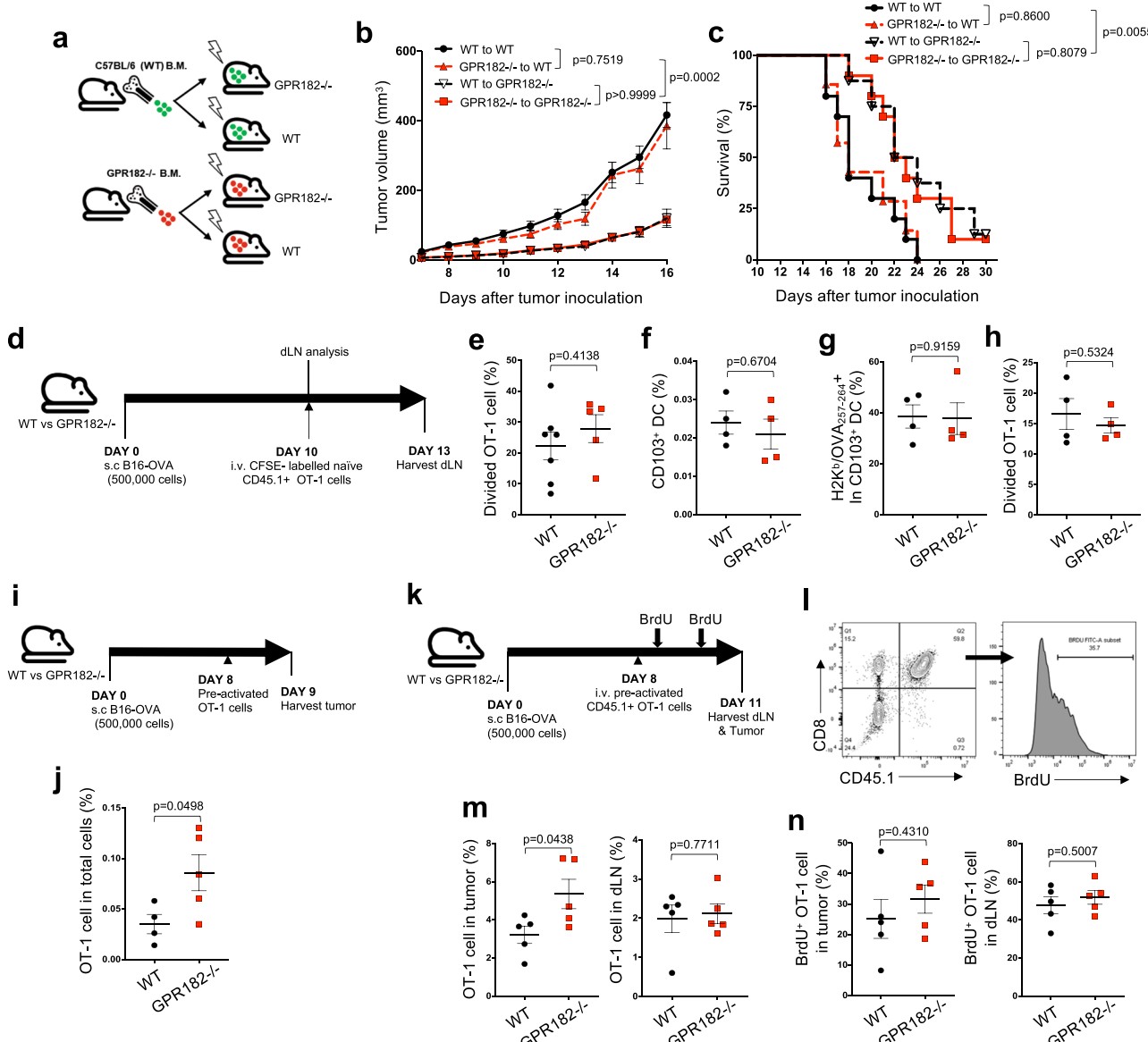

**Fig. 4 Improved homing, not priming, contributes to TIL accumulation in GPR182−/− mice. a–c** Schematic of generating GPR182−/− bone marrow chimeras (**a**). Tumor growth curves (**b**) and Kaplan–Meier curves (**c**) for chimera mice inoculated subcutaneously with B16 tumor cells were recorded. Pooled data from two independent experiments. WT to WT, n = 10; GPR182−/− to WT, $n = 7$; WT to GPR182−/−, n = 8; GPR182−/− to GPR182−/−, $n = 10$. **d–h** Schematic of dLN T cell priming (**d**). Divided OT-1 cells in tumor dLN were quantified 72 h after transfer into mice bearing B16-OVA tumors (WT, $n = 7$; GPR182−/−, $n = 5$) (**e**). **f–h** After 10 days of tumor inoculation, single-cell suspensions were prepared from tumor dLN. The frequencies of CD103 + DC (**f**) and H2K$^b$/OVA$_{257-264}$-positive cells in CD103 + DC population (**g**) were quantified by flow cytometry. Cell suspensions of dLN were used to stimulate CFSE-labeled naïve OT-1 cells, and divided OT-1 cells after 3-days co-culture were quantified. $n = 4$ per group (**h**). Representative data from two independent experiments. **i, j** Schematic of pre-activated OT-1 cell transfer (**i**). The frequencies of transferred OT-1 cells in tumors were quantified 24 h after transfer (**j**). $n = 4$ and 5 per group, Representative data from two independent experiments. **k–n** Schematic of pre-activated OT-1 cell homing (**k**). Representative flow cytometry plot of transferred OT-1 T cells (CD45.1 + CD8+) and BrdU staining (**l**). 3 days after transfer, the frequencies of transferred OT-1 cells (CD45.1 + CD8+) (**m**) and BrdU+ OT-1 cells (**n**) were quantified in tumors and dLN from WT and GPR182−/− mice. $n = 5$ per group. Representative data from two independent experiments. $P$ values from two-sided student's $t$-tests (**e–n**). Error bars represent SEM.

surface binding between FLAG-tagged human CXCL9 (hCXCL9-FLAG) and CHO cells stably expressing human GPR182 (GPR182 + CHO). After incubated for 30 min at 4 °C, hCXCL9-FLAG strongly bound to GPR182 + CHO cells but not WT CHO cells; this interaction could be abolished by pre-incubating cells with excess untagged human CXCL9 (Fig. 6a). Competition assay with titrated unlabeled CXCL9 further revealed that the $K_d$ of CXCL9 is at $1.33 \times 10^{-6}$ (M) with an IC50 of $3.03 \times 10^{-6}$ (M) (Fig. 6b). Mouse CXCL9 protein was able to bind HEK293T cells transiently expressing mouse GPR182, verifying that this

interaction was conserved in mouse (Fig. 6c). We assessed indirectly whether GPR182 can also bind CXCL10 and CXCL11, the other two CXCR3 ligands. The inclusions of titrated CXCL10 or CXCL11 were able to strongly compete with CXCL9 for the binding of GPR182 + CHO cells (Fig. 6d), with IC50 of $1.47 \times 10^{-7}$ (M) and $6.87 \times 10^{-7}$ (M), respectively. This indirectly demonstrates that GPR182 interacts with the other two chemokine ligands for CXCR3. Interestingly, GPR182 shares the closest homology to ACKR3 which happens to interact with CXCL11[36], another chemokine ligand for CXCR3.

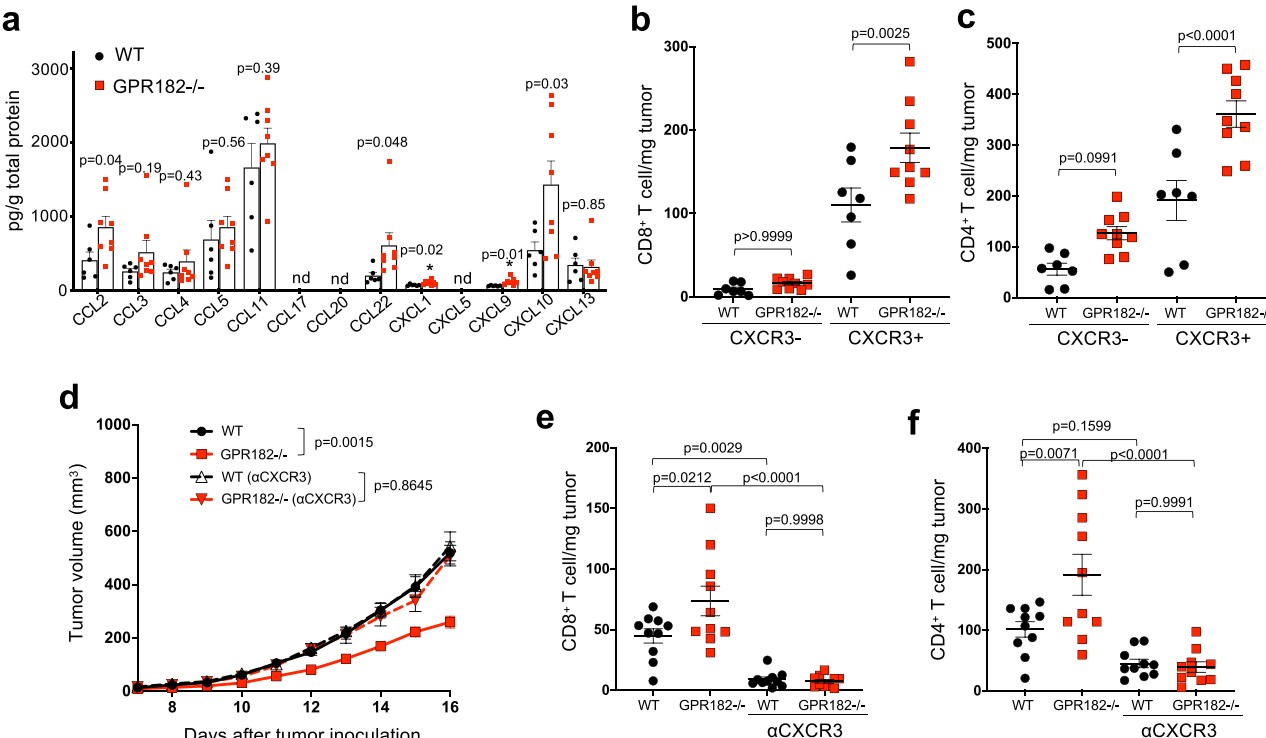

**Fig. 5 The CXCR3 pathway mediates increased TIL infiltration in GPR182−/− mice. a–c** 16 days after tumor inoculation, YUMM1.7 tumors in WT and GPR182−/− mice were collected for analyses. Intratumoral chemokine concentrations in YUMM1.7 tumors were measured using LEGENDplex ProInflammatory Chemokine Panel (**a**). WT, $n = 6$; GPR182−/−, $n = 8$. The densities of CXCR3- and CXCR3+ cells in CD8+ (**b**) and CD4+ (**c**) T cells within YUMM1.7 tumors were quantified by flow cytometry. WT, $n = 7$, GPR182−/−, $n = 9$. Representative data from two independent experiments. **d–f** Tumor growth curves of WT and GPR182−/− mice that were inoculated with 50,000 YUMM1.7 tumor cells and were followed with or without the treatment of CXCR3 mAb (**d**). Flow cytometric quantification of CD8+ (**e**) and CD4+ T cell (**f**) densities in WT and GPR182−/− mice 16 days after tumor inoculation. $n = 10$ per group. Representative data from two independent experiments. P-values from two-sided Students t-test (**a**), one-way ANOVA with Bonferonni correction for multiple comparisons (**b**, **c**, **e**, **f**), and two-way ANOVA test (**d**). Error bars represent SEM.

ACKRs are known to internalize chemokines upon receptor binding[37–41]. In primary human LECs, we found that GPR182 protein was mainly present intracellularly (Supplementary Fig. 5d). Treatment of LECs with DMOG led to significant upregulation of GPR182, supporting that GPR182 can be induced by hypoxia[42]. When we incubated human LECs with GPR182 mAb at 37 °C over one hour, we were able to detect GPR182 antibody endocytosis (Supplementary Fig. 5e). We were unable to detect CXCL9 endocytosis by primary LECs, which might be due to the weak interaction and low GPR182 expression. Therefore, we use GPR182 + CHO transfectant to evaluate the capacity of GPR182 to mediate chemokine ligand internalization. WT and GPR182 + CHO cells were incubated with AF488-labeled CXCL9 at 37 °C for 20 min. When we examined CHO cells by confocal microscopy, AF488-CXCL9 was detected intracellularly in GPR182 + CHO cells, but not in WT CHO cells (Fig. 6e). At the same time, surface GPR182 was internalized upon incubation with human CXCL9 (Fig. 6f). This suggested that GPR182 interacts with and endocytoses CXCL9. We further stained GPR182 together with markers of endocytosis, endosomes, and lysosomes to determine the intracellular fate of GPR182 after CXCL9 incubation. Intracellular GPR182 co-localized with clathrin, EEA1, and LAMP1 while there was no co-localization with caveolin-1 (Fig. 6g–j). This indicates that GPR182 undergoes clathrin-mediated endocytosis upon binding and traffics to early EEA1-positive endosomes and LAMP1-positive lysosomes[43]. In human melanoma, CXCL9, expressed by macrophages and DCs[10,44], is primarily present in the peritumoral stroma where lymphatic GPR182 locates (Supplementary Fig. 5f). This close

spatial relationship supports the possibility of CXCL9 scavenging by GPR182 in the TME. Consistently, in mouse melanoma models, GPR182-expressing LECs are mainly present in the periphery of the tumors, where intratumoral CD3+ T cells primarily locate (Supplementary Fig. 5g). Taken together, our studies demonstrate GPR182 acts as an ACKR to scavenge chemokines ligands for CXCR3 and target them for intracellular degradation.

**GPR182 interacts with chemokines promiscuously via the GAG-binding motif.** ACKRs often interact with multiple chemokines[13,45]. Using a competitive binding assay, we further analyzed the abilities of a panel of 35 human chemokines to block the interaction between GPR182 and CXCL9. Chemokines from all four subclasses (CCL, CXCL, CX3CL, and XCL) were able to block the interaction between CXCL9-AF488 and GPR182 + CHO cells to varying degrees, suggesting that GPR182 binds chemokines promiscuously (Fig. 7a). Because chemokines are known to possess GAG-binding motifs to attach to the endothelium[7], we hypothesized that GPR182 is a pattern-recognition receptor for the GAG-binding motif. Supporting that, pre-incubation of GPR182 + CHO cells with a C-terminal GAG-binding peptide from human CXCL9 (69-93)[46] completely blocked the interaction between GPR182 and CXCL9 (Fig. 7b). When we synthesized two generic GAG-binding peptides consisting of repeats of GAG-binding motifs [$(ARKKAAKA)_3$ and $(AKKARA)_4$][47], these two peptides were able to markedly disrupt the GPR182 + CHO cell binding by CXCL9 (Fig. 7c). Thus, GPR182 may bind a broad array of GAG-binding proteins.

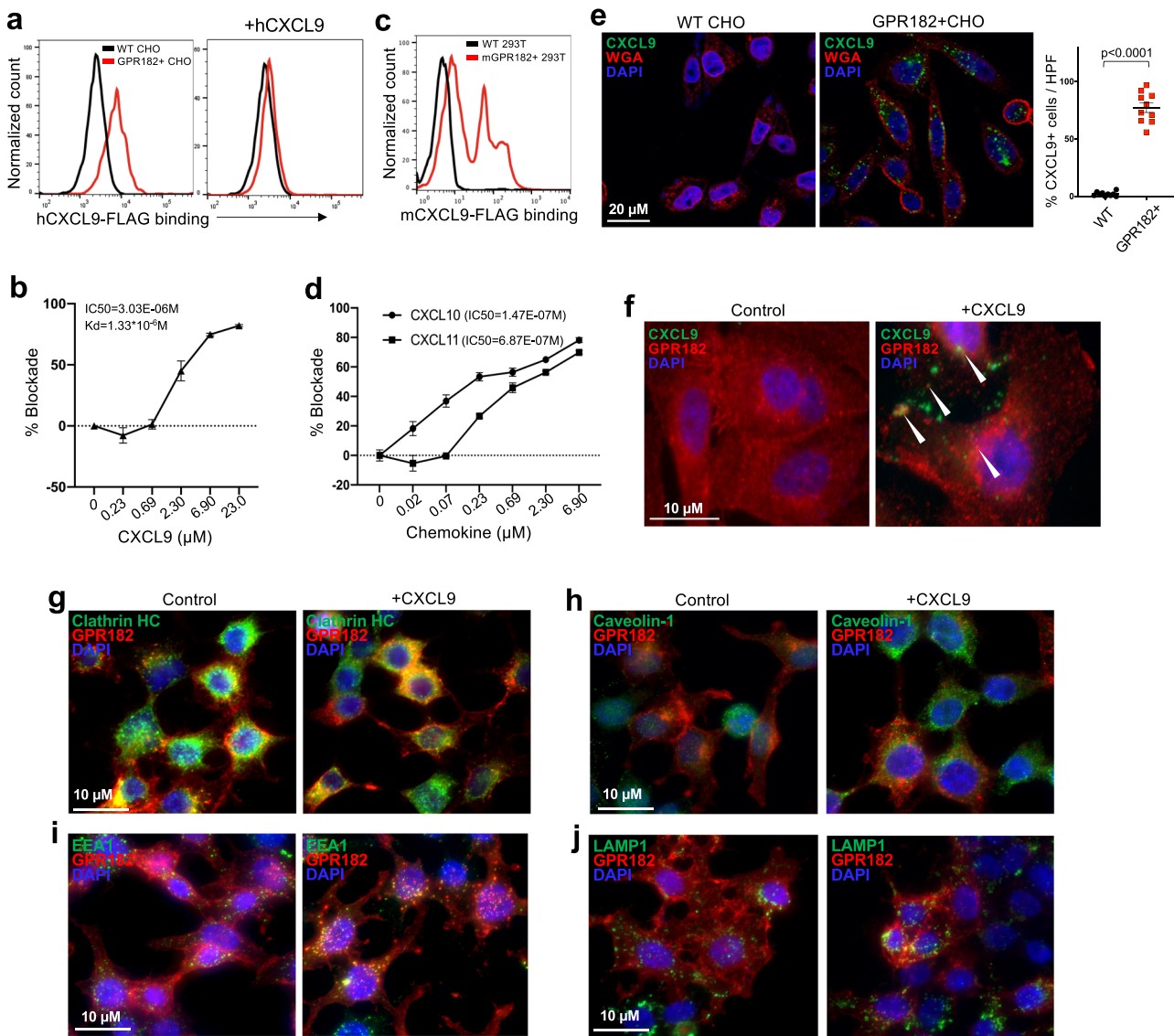

**Fig. 6 GPR182 binds and endocytoses chemokine CXCR3 ligands. a, b** WT CHO cells and CHO cells stably transfected with human GPR182 (GPR182 + CHO) were stained for human CXCL9-FLAG binding at 4 degrees. Cells were pre-incubating with purified human CXCL9 to assess its blocking capacity (**a**). The addition of different concentrations of unlabeled CXCL9 was used to quantify the IC50 and Kd (**b**). *n* = 3 replicate samples. Representative data from at least three independent experiments. **c** WT 293T cells and 293T cells transiently transfected with mouse GPR182 were stained for mouse CXCL9-FLAG binding at 4 degree. *n* = 3. Representative data from two independent experiments. **d** GPR182 + CHO cells were stained for hCXCL9-AF488 at 4 degree. Cells were pre-incubated with different concentrations of CXCL10 or CXCL11 to assess their blocking capacities. *n* = 3 replicate samples. Representative data from two independent experiments. **e, f** WT and GPR182 + CHO cells were incubated with CXCL9-AF488 at 37 degrees for 20 min, followed with fixation and costaining with wheat-germ agglutinin (WGA). CXCL9-AF488 after incubation with WT or GPR182 + CHO cells was determined by confocal microscopy. *n* = 10 per group (**e**). The location of GPR182 with or without CXCL9 incubation was determined by confocal microscopy (**f**). Representative data from two independent experiments. **g–j** WT and GPR182 + 293T cells were incubated with CXCL9 for 20 min at 37 degrees followed with fixation and staining for GPR182 and Clathrin HC (**g**), Caveolin-1 (**h**), EEA1 (**i**), or LAMP1 (**j**) before confocal microscopy. Representative data from two independent experiments. *P*-value from two-sided paired *t*-test (**e**). Error bars represent SEM.

We further assessed the signaling pathways triggered by GPR182, in comparison to CXCR3. Upon the stimulation of CXCL11, the most potent CXCR3 agonist[48], we observed a strong calcium flux in CXCR3 + CHO cells, but not GPR182 + cells (Supplementary Fig. 5h). Consistently, western blot analysis indicated that CXCL11 treatment triggered ERK phosphorylation in CXCR3 + CHO cells, but not GPR182-expressing cells (Supplementary Fig. 5i). Besides canonical G protein-mediated signaling, GPCRs differentially recruit β-arrestin to coordinate their signaling[49]. We used nanoluciferase binary (NanoBiT) technology[50] to examine the association between β-arrestin and

GPR182. In this system, we made expression constructs in which GPR182 and CXCR3 were fused to the N-terminal of a small 11-aa complementing peptide (smBiT) while β-arrestin2 was added to the C-end of the complimentary large BiT protein (LgBiT). The association between these two GPCRs and β-arrestin2 would lead to the activation of luciferase (Supplementary Fig. 5j, left). We observed a strong luminescent signal in HEK293T cells which were co-transfected with GPR182 and β-arrestin2, demonstrating that GPR182 was constitutively associated with β-arrestin (Supplementary Fig. 5j, right). These data suggest that GPR182 does not trigger classical G-protein signaling, but is constitutively

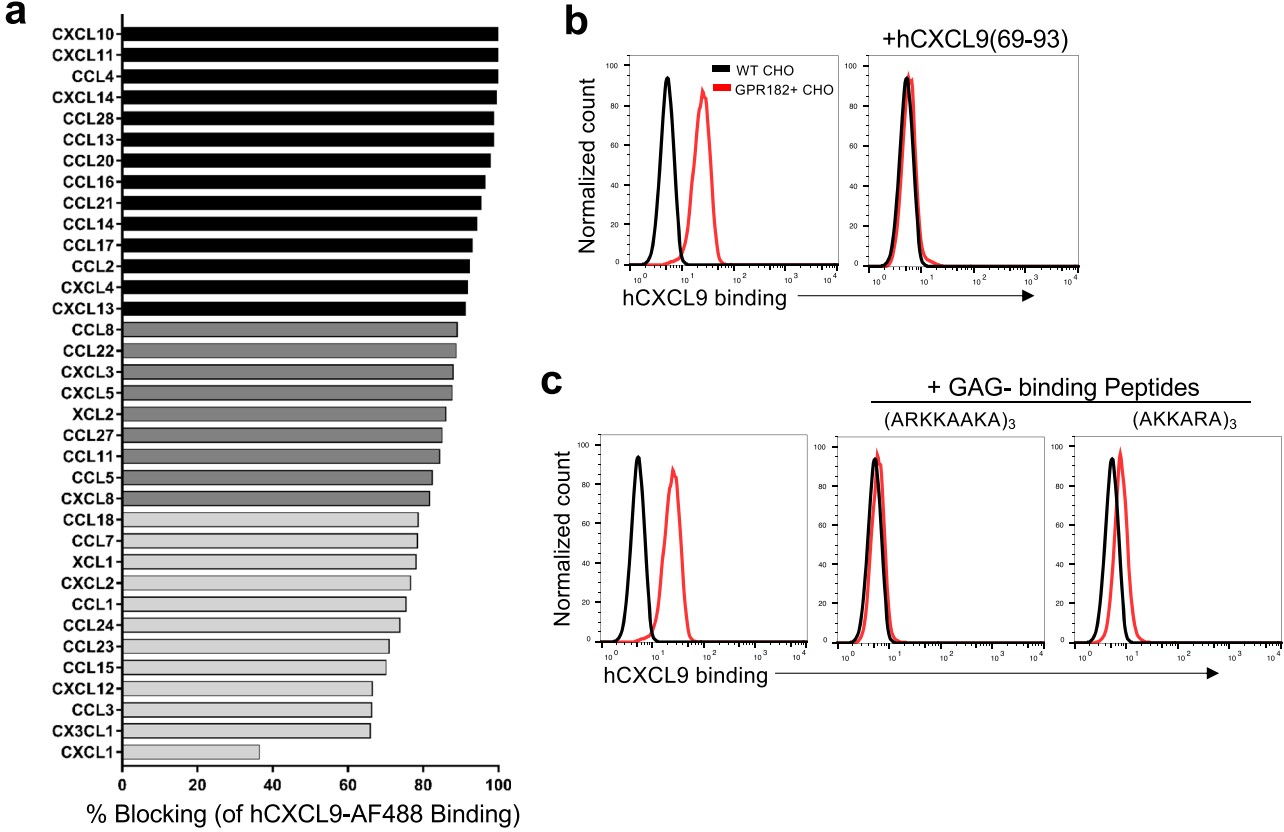

**Fig. 7 GPR182 promiscuously binds chemokines via the GAG-binding motif. a** A panel of 35 human chemokines were tested for their ability to block the binding GPR182 + CHO cells by hCXCL9-AF488. % blocking is shown for all chemokines, with chemokines that blocked strongly shown in black bars (>90% blocking), moderately in dark gray (80–90% blocking), and weakly in light gray (<80% blocking). Representative data from two independent experiments. **b** WT and GPR182 + CHO cells were stained for hCXCL9-AF488 binding at 4 degree. Cells were pre-incubated with a 25-aa peptide of CXCL9 (CXCL9(69-93)) to assess its blocking capacity. Representative data from at least three independent experiments. **c** Two synthetic GAG-binding peptides were assessed for their ability to block the binding of GPR182 + CHO cells by CXCL9-AF488. Representative data from two independent experiments.

associated with β-arrestin, which is consistent with a recent publication[51].

**GPR182 ablation sensitizes immunologically cold melanoma to immunotherapy.** YUMM1.7 tumors are a mouse melanoma cell line with poor immunogenicity and are resistant to ICB therapy[29,52]. Given the increased T cell infiltration and functions in YUMM1.7 tumors from GPR182−/− mice, we hypothesized that GPR182 ablation would sensitize YUMM1.7 to ICB therapy. Eight days after tumor inoculation, mice were treated with the dual ICB therapy (anti-CTLA4 and anti-PD1 antibodies). We observed no significant reduction in YUMM1.7 tumor growth or overall survival in WT mice treated with the dual ICB therapy, which is consistent with previous observation[29]. However, GPR182−/− mice exhibited a significant reduction in tumor growth with ICB therapy (Fig. 8a); as a result, ICB further extended the overall survival of GPR182−/− mice (Fig. 8b). In addition, B16 tumors, another immunologically cold mouse melanoma, exhibited increased responsiveness to single-agent anti-PD1 in GPR182−/− mice (Fig. 8c). In WT mice, anti-PD1 therapy alone had no effect on B16 tumor progression while the same treatment in GPR182−/− mice was able to significantly slow tumor growth (Fig. 8c).

We further evaluated the efficacy of adoptive T cell therapy in GPR182−/− mice. WT and GPR182−/− mice were inoculated with B16-OVA tumors at an inoculum in which tumor outgrowth was similar between two groups. Following tumor engraftment, mice received an intravenous transfer of 5 million in vitro activated OT-1 cells on day 6 after tumor inoculation. The infusion of OT-1 cytotoxic T lymphocytes (CTL) alone had no effect on inhibiting tumor progression in WT mice, however, GPR182−/− mice that received the same number of OT-1 CTL displayed a significant attenuation in tumor growth (Fig. 8d). Taken together, our data demonstrate that GPR182 functions as a chemokine scavenger to limit effector T cell infiltration in the TME (Fig. 8e), and GPR182 ablation improves responsiveness to ICB therapy and promotes the efficacy of ACT in poorly immunogenic tumors.

**Discussion**
Our studies demonstrated that GPR182 deficiency leads to attenuated tumor growth that is T cell-dependent in multiple models of murine melanoma and identifies GPR182 as a potential therapeutic target to improve the effectiveness of immunotherapy. Loss of GPR182 led to increased intratumoral levels of chemokines, which triggered a CXCR3-dependent increase in infiltration of effector CD4+ and CD8+ T cells. Furthermore, we demonstrated that GPR182 acts as an ACKR to scavenge chemokines broadly.

To date, the family of ACKRs includes at least four members, ACKR1 (DARC), ACKR2 (D6), ACKR3 (CXCR7), and ACKR4 (CCRL1 or CCX-CKR)[53]. ACKRs share homology with conventional chemokine receptors, but differ significantly in tissue expression, signaling, and function. Chemokines are key regulators

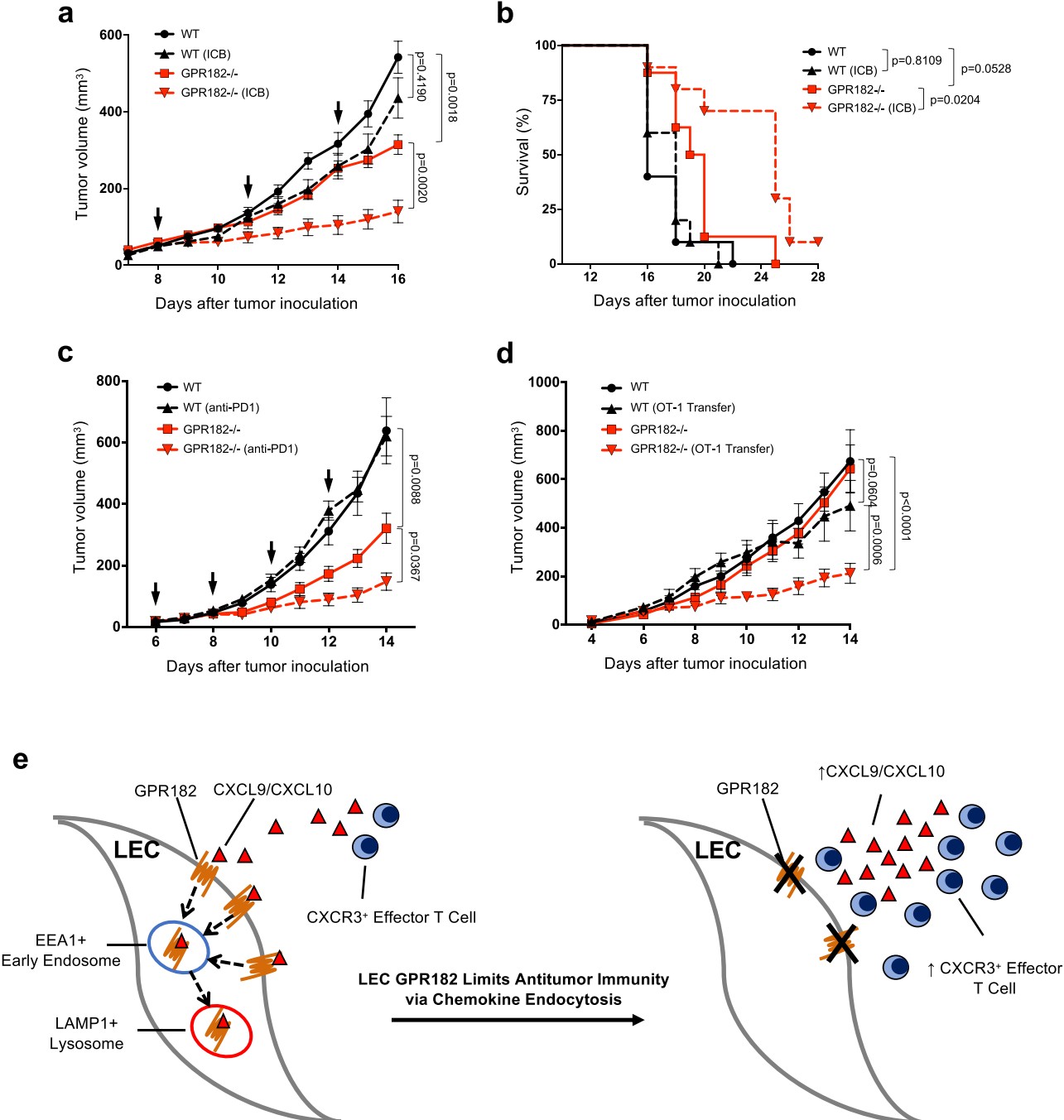

**Fig. 8 GPR182 ablation sensitizes tumors to immunotherapy. a**, **b** WT and GPR182−/− mice were inoculated with 50,000 YUMM1.7 tumor cells and treated with ICB (anti-CTLA-4 and anti-PD1) or vehicle control (arrows indicate ICB treatment). Tumor growth (**a**) and Kaplan–Meier survival (**b**) curves for YUMM1.7 were indicated. WT, $n = 10$; WT (ICB), $n = 10$; GPR182−/−, $n = 8$; GPR182−/− (ICB), $n = 10$. **c** Tumor growth curves for WT and GPR182−/− mice inoculated with B16 tumors and treated with anti-PD1 or control after the establishment of tumors (arrows indicate anti-PD1 therapy). $n = 10$ per group. **d** WT and GPR182−/− mice were s.c. inoculated with 500,000 B16-OVA tumor cells; On day 6, half of the mice received i.v. transfer of activated OT-1 cells. Tumor growth curves for B16-OVA were followed daily. $n = 10$ per group. Representative data from two independent experiments. **e** A scheme describes how GPR182 functions as a chemokine scavenger to suppress antitumor immunity. *P*-values for growth curves from two-way ANOVA test (**a**, **c**, **d**). *P*-values for tumor mass from student's *t*-test. *P*-values for Kaplan–Meier curves from log-rank test, two-sided (**b**). Error bars represent SEM.

of leukocyte migration through activating conventional chemokine receptors expressed on distinct leukocyte populations[54]. In contrast, ACKRs are expressed on stromal cell populations, including lymphatic and blood ECs, and are unable to activate classical G-protein-mediated signaling due to mutations in the DRYLAIV motif[13–15]. As opposed to directly promoting leukocyte migration, ACKRs act to modulate inflammation by shaping chemokine

gradients in tissues through chemokine binding, degradation, and transcytosis[13–15]. Our data strongly implicates GPR182 as an additional member of the ACKR family. GPR182 shares close homology with the chemokine receptor family[18], but has mutations in the DRYLAIV motif ("DRYVTLT" in GPR182). Consistently, GPR182 does not trigger canonical G protein-mediated signaling while it is constitutively associated with β-arrestin. Our

data and other publications[21,22,55,56] indicate that GPR182 is widely expressed in ECs, particularly LECs and sinusoidal ECs, but not leukocytes. In the TME, we demonstrate that GPR182 is expressed specifically on LECs, but whether GPR182 is expressed on vascular EC or other stromal cell populations in tumors has not been fully excluded. GPR182 broadly interacts with human chemokines via the GAG-binding motif. Upon receptor binding, chemokines are actively endocytosed by GPR182 and targeted to the intracellular early endosomes and lysosomes. Together these data support GPR182 functions as a chemokine scavenger (Supplementary Table 1).

Our data are in agreement with a recent report by Le Mercier et al.[19], which demonstrated that GPR182 acts as an ACKR with the highest affinity for CXCL10, CXCL12, and CXCL13. Phenotypically, GPR182 deficient mice had increased plasma concentrations of CXCL10, CXCL12, and CXCL13, as well as decreased retention of hematopoietic stem cells in the bone marrow. We have further built on these findings by demonstrating the role of GPR182 scavenging of CXCL9 and CXCL10 by LECs in the TME of murine melanoma, implicating lymphatic GPR182 as a potential therapeutic target in cancer immunotherapy. In addition, we show that GPR182 interacts with the GAG-binding motif present across chemokines, providing an explanation for the promiscuous chemokine binding capacity of GPR182. We also demonstrate a broader chemokine binding profile of GPR182 than reported by Le Mercier et al.[19]. One possible reason that Le Mercier et al. found only a limited chemokine binding profile for GPR182 is that they used a lower concentration of chemokines (120 nM) to compete for GPR182 binding with CXCL10. However, in a similar assay, we used a much higher concentration of chemokines (20 µg/ml, about 2.4 µM) for binding competition with CXCL9, which has a lower affinity for GPR182 than CXCL10.

In human malignancies, tumor lymphangiogenesis is often associated with increased lymph node metastases and poor prognosis[57,58]. Cancer cells utilize lymphatics as a conduit to metastasize to regional lymph nodes and this is uniformly a poor prognostic marker. Lymphatics and LECs may also enable cancer to evade the immune system through tolerance induction[59]. Besides regulating antigen presentation and DC homing, recent studies revealed that lymphatics actively modulate antitumor immunity, including PD-L1 upregulation[60,61] and the recruitment of immune cells into the TME[62]. Despite LECs being primarily located in the tumor periphery they actively help to shape the concentration of inflammatory cytokines in the TME and influence T cell infiltration[63]. Many ACKRs are found to be expressed in lymphatics and many of them are proven to modulate multiple chemokines to affect the TME[17]. Our finding that GPR182 functions as a new ACKR to broadly endocytose chemokines provides new molecular evidence for the immunomodulatory role of the lymphatics. By clearing proinflammatory chemokines via GPR182, LECs actively retain immune responses to maintain tissue tolerance.

The chemokine system is often described as redundant or promiscuous due to overlapping interactions between chemokines and their receptors[64]. Despite the complex and overlapping interactions between chemokines and their receptors, the biologic significance of these interactions is not uniform. Different chemokines can interact with the same receptor to induce different degrees of receptor internalization, receptor recycling, and ligand degradation. Furthermore, the tissue- and cell type-specific expression patterns and temporal expression dynamics affect the biologic relevance of these specific interactions[64]. Our in vivo observations led us to discovering that GPR182 interacts with CXCL9, CXCL10, and CXCL11 and, in turn, modulates the CXCR3 axis to limit T cell homing into the TME. As CXCL9 and CXCL10 are the only functional ligands for CXCR3 in C57BL/6 mice, complete reversal of the anti-tumor phenotype and improved T cell infiltration in GPR182-deficient mice with CXCR3 blockade demonstrates that alteration in the abundance of these two chemokines in the TME is driving the observed phenotype. We also observed increased concentrations of CCL2, CCL22, and CXCL1 in tumors of GPR182-deficient mice; these findings allowed us to further discover that GPR182 broadly interacts with chemokines via their GAG-binding motif. CCL2, CCL22, and CXCL1 interact with CCR2, CCR4, and CXCR2, respectively to mediate chemotaxis of monocytes, macrophages, DCs and T cells[14]. While we did not find significant differences in macrophage or DC density in tumors from GPR182-deficient mice, we did observe increased PMN-MDSC concentrations in GPR182-deficient mice which may be secondary to increased CCL2, CCL22 and CXCL1 concentrations. However, the increased PMN-MDSC concentrations were modest in comparison to the T cell phenotype we observed. It remains to be determined why GPR182 regulates certain chemokines but not others in our mouse tumor models. Furthermore, it is possible that increased inflammatory chemokines in GPR182−/− tumors could be a secondary effect of enhanced inflammation in the TME. On the other hand, the dominant role of the CXCR3 pathway in regulating effector T cell infiltration in melanoma might mask the contributions of other chemokines which GPR182 regulates. In addition, GPR182 is the only proposed ACKR with a known affinity for CXCL9 and CXCL10. The lack of redundancy in scavenging receptors for CXCL9 and CXCL10 may allow for ablation of GPR182 to have a more dominant effect on these chemokines. We further demonstrated that GPR182 is a pattern-recognition receptor for GAG-binding motifs. Therefore, it is very likely that GPR182 has a broader biological function beyond the immune system[20,65]. Future research into additional GAG-binding proteins possibly regulated by GPR182 and the consequences is warranted.

CXCR3 and its three chemokine ligands, CXCL9, CXCL10, and CXCL11 play a central role in regulating effector T cell homing. CXCR3 expression is induced on naïve T cells after activation and is highly expressed on effector CD8+ T cells and Th1-type CD4+ T cells[66]. B6 mice produced a non-functional form of CXCL11[67], but CXCL9 and CXCL10 have both been demonstrated to play significant roles in mediating T cell infiltration into murine tumors, with CXCL9 appearing to play a dominant role[10,68,69]. Upon CXCR3-dependent T cell infiltration into the TME or other sites of inflammation, a chemokine-dependent positive feedback loop exists in which the recruitment of CXCR3-positive T cells leads to increased production of IFN-γ- inducible chemokines, and in turn, recruits more CXCR3- positive T cells[66]. This positive feedback loop provides an additional explanation for the significant increase of T cell infiltration and enhanced antitumor immunity caused by GPR182 ablation. On the other hand, TILs emigrate from primary tumors to dLNs via afferent lymphatic vessels[70], therefore, it is possible that lymphatic GPR182 limits TIL infiltration by regulating T cell retention in the tumors. As CXCR3 is expressed on many melanomas[71], increased chemokines by GPR182 ablation might have a direct impact on tumor cells. The scarcity of proinflammatory chemokines in immunologically cold tumors would suggest that GPR182 ablation could be most beneficial in these tumors. In two immunologically cold melanoma tumor models, we were able to demonstrate that GPR182 ablation alone recruited sufficient effector T cells to slow tumor progression, and at the same time sensitized otherwise resistant tumors to subsequent immunotherapy. It remains to be determined if a therapeutic agent that blocks GPR182 mediated chemokine scavenging can deliver an antitumor immunity comparable to GPR182 knockout mice in these ICB-unresponsive tumors.

GPR182 expression has been previously studied in the context of intestinal regeneration following irradiation and adenoma formation. Kechele et al. identified GPR182 expression to be enriched in crypt base columnar intestinal stem cells. Loss of GPR182 had no effect on homeostatic intestinal proliferation but was associated with intestinal hyperproliferation following irradiation injury and increased adenoma burden in the APC $^{(min/+)}$ mouse model. The authors conclude the GPR182 acts to suppress intestinal proliferation and therefore may suppress tumorigenesis[22]. In comparison to this prior work, we demonstrate an anti-tumor effect with loss of GPR182. Important differences between these studies include a cell-intrinsic effect of GPR182 on intestinal stem cells versus a cancer cell extrinsic effect of stromal cells scavenging chemokines in the TME. In human melanoma and breast cancer samples, GPR182 expression was only detected on LECs, making a cancer cell extrinsic effect more biologically relevant.

In conclusion, we identify GPR182 as an ACKR that negatively regulates antitumor immunity in melanoma through chemokine scavenging. Therapeutic agents targeting GPR182 will have significant translational potential as an approach towards improving T cell infiltration into the TME of immunologically cold tumors, so as to expand the effectiveness of immunotherapy.

## Methods

**Plasmids and cell line transfection**. Human and mouse GPR182 cDNAs were cloned by PCR and the full-length coding regions were further put into a pcDNA3.1(−) expression vector by restricted digestion. FLAG-mouse CXCL9 and FLAG-human CXCL9 were cloned by PCR and further constructed into a pFLAG-CMV expression vector. Lipofectamine 3000 kit (Invitrogen) was used to perform transfection on HEK293T and CHO cells according to the manufacturer protocol. Stably transfected cell lines were isolated following limiting dilution.

**Mice**. All mice were housed at the University of Colorado animal facility under pathogen-free conditions and all animal care procedures and experiments were approved by the Institutional Animal and Use Committee at the University of Colorado Anschutz Medical Campus (Aurora, CO). GPR182−/− mice were bred in-house using *Gpr182*$^{tm2a(KOMP)Wtsi/+}$ (knockout first/promoter driven) mice which were generated and obtained from the KOMP Repository (www.komp.org) and backcrossed for five generations with C57BL/6J mice. *Gpr182*$^{tm2a(KOMP)Wtsi/+}$ were crossed to achieve homozygous GPR182$^{lacZ/lacZ}$ mice as previously described[22]. *GPR182*$^{lacZ/lacZ}$ mice have negligible expression of endogenous GPR182 compared to GPR182$^{+/+}$ controls[22]. Genotyping was performed with the following primers: CSD-neoF: GGATCTCATGCTGGAGTTCTTCG; CSD-GPR182-ttr: GTACCCA ACAAGGTTTCTTCCCAGC; CSD-GPR182-F2: GACC AAATAGCAAGGCAGAACAGG. C57BL/6 mice (stock# 000664) at 6 to 8 weeks of age were purchased from the Jackson Laboratory (Bar Harbor, ME). C57BL/6 mice were used as wild-type controls, unless otherwise specified where wild-type littermates were used as controls. OT-1 T cell receptor (TCR) transgenic mice (stock# 003831) were obtained from the Jackson Laboratory and crossed with CD45.1+/+ mice to obtain CD45.1+ OT-1 mice. Experiments used age and sex-matched mice between 8 and 12 weeks of age. Experimental and control mice were co-housed for at least 2 weeks before the start of the experiments.

**Tumor models**. Three murine melanoma tumor cell lines were used. B16 melanoma cells were originally obtained from ATCC (ATCC.org). B16-OVA cells were acquired from Dr. Lieping Chen (Yale University, New Haven, CT). YUMM1.7 and YUMMER1.7 cells were acquired from Dr. Mayumi Fujita (University of Colorado, Aurora, CO). B16, B16-OVA, and YUMM1.7 cells were cultured in RPMI-1640 media containing 10% fetal bovine serum, 2% HEPES buffer, and 1% penicillin/streptomycin. YUMMER1.7 cells were cultured in DME-F12 media containing 10% fetal bovine serum and 1% penicillin/streptomycin. B16 cells (1 × 10$^5$ or 2 × 10$^5$), B16-OVA cells (5 × 10$^5$), YUMM1.7 cells (5 × 10$^4$) or YUMMER1.7 (2.5 × 10$^5$ cells/mouse) were injected subcutaneously on the right hind flank in 100 μL HBSS. Both male and female mice were used for B16 and B16-OVA tumor inoculation while only male mice were used for YUMM1.7 and YUMMER1.7 tumors. When tumors became palpable, they were measured every 1-2 days using digital calipers, with tumor volume calculated as 1/2 x(longest dimension × perpendicular dimension$^2$). For survival studies, end point was death, significant tumor ulceration requiring euthanasia, or tumor volume >1000 mm$^3$ (YUMM 1.7 and YUMMER1.7) or 500 mm$^3$ (B16). The method of euthanasia was performed in accordance with the US Department of Health and Human Services Guide for the Care and Use of Laboratory Animal. Carbon dioxide was

administered for 2 min from a compressed gas tank. Cervical dislocation was added as a secondary physical euthanasia.

### In vivo treatments

*CD4/CD8 depletion*. To deplete CD4+ and CD8+ T cell populations, mice were injected intraperitoneally with anti-CD4 (clone GK1.5) and anti-CD8β (clone 53-5.8) (Bio X Cell; Lebanon NH) at a dose of 300 μg per mouse 1 day prior to tumor inoculation and at a dose of 200 μg on day 7 after tumor inoculation. CD4 and CD8 T cell depletion was confirmed from peripheral blood by flow cytometry.

*OT-1 T cell transfer and BrdU labeling*. Single-cell suspensions were prepared from the spleen and lymph nodes of CD45.1+ OT-1 TCR transgenic mice. Following ACK lyses of red blood cells, OT-1 cells were cultured for 48 h in RPMI-1640 media supplemented with 10% fetal bovine serum and 10 ng/mL SIINFEKL peptide (GenScript) at a cell density of 1–2 × 10$^6$ cells per mL. After 48 h of activation, media was replaced with RPMI-1640 supplemented with 10% fetal bovine serum and 100 U/mL IL-2 (Sigma-Aldrich) for an additional 48 h. Prior to the intravenous transfer, viable OT-1 cells were isolated via Ficoll density gradient separation. For in vivo BrdU labeling of transferred OT-1 cells, mice were injected intraperitoneally with 1 mg (0.1 mg/mL) of BrdU (BD Bioscience) in 1× PBS 24 and 48 h after OT-1 transfer.

*CXCR3 blockade*. To block CXCR3, mice were injected intraperitoneally with anti-CXCR3 (clone CD183; Bio X Cell; Lebanon NH) at a dose of 200 μg per mouse 1 day prior to tumor inoculation and 7 days after tumor inoculation.

*Immune Checkpoint blocker and Adoptive T Cell Transfer Therapy*. For combinatory anti-PD1 and anti-CTLA-4 therapy in YUMM1.7 tumors, treatment was initiated 8 days after tumor inoculation when all tumors were measurable. Mice were treated with 150 μg of anti-mPD-1 (clone RMP1-14; Bio X Cell; Lebanon NH) and 150 μg of anti-mCTLA-4 (clone 9D9; Bio X Cell; Lebanon NH) intraperitoneally or with vehicle control (1× PBS) every three days for a total of three injections. For single-agent anti-PD1 therapy in B16 tumors, treatment was started 6 days after tumor inoculation when all tumors were measurable. Mice were treated with 200 μg of anti-mPD-1 (clone RMP1-14) intraperitoneally or with vehicle control (1× PBS) every other day for a total of four treatments. For adoptive T cell transfer therapy, OT-1 cells were activated in vitro, as described above, and 5 million cells were transferred intravenously into tumor-bearing mice 6 days after B16-OVA tumor inoculation.

**Single-cell isolation from tissue**. Tumors tissue were resected and weighed before processing. Following mechanical homogenization, the tissue was resuspended in RPMI1640 media containing Liberase DM (Roche Diagnostics Corporation). Tumors were digested for 30 min at 37 °C and subsequently passed through a 100 μm cell strainer. For isolation of single-cell suspensions from lymph nodes, tissue was first disrupted with 18-gauge needles prior to digestion.

**Flow cytometry analysis**. Single-cell suspensions were blocked with LEAF anti-mouse CD16/32 (anti-FcγIII/II receptor, clone 93; BioLegend) for 20 min before staining with primary conjugated antibodies. Staining antibodies were purchased from BioLegend, unless otherwise specified, and included: CD45.2 (104), CD19 (6D5), CD3 (17A2), NK1.1 (PK136), CD11b (M1/70), CD11c (N418), I-Ab (AF6-120.1), Ly-6G (IA8), Ly-6C (HK1.4), F4/80 (BM8), CD8a (53-6.7), CD4 (RM4-5), CD44 (IM-7), CD62L (MEL-14), PD1 (RMPI-30), FoxP3 (MF-14), H-2Kb bound SIINFEKL (25-D1.16), GzmB (GB11), IFN-γ (XMG1.2), PD1 (RMPI-30), 4-1BB (17B5), CXCR3 (CXCR3-173) and Ghost Dye Red 780 (13-0865-T100, Tonbo Biosciences). BrdU staining kit was purchased from BD Bioscience. For GzmB and IFN-γ staining, single-cell suspensions were stimulated ex vivo in the presence of Monensin solution (BioLegend). Stimulated and un-stimulated control splenocytes were used as gating controls for cytokine staining. For intracellular staining, surface antigens were stained prior to fixation and permeabilization with eBioscience FoxP3/Transcription Factor Staining Buffer Set (Thermo Fisher Scientific). Flow cytometric analysis was conducted on BD FACS Calibur and Beckman Coulter CytoFlex S and data were analyzed by FlowJo software (FlowJo, LLC).

**Bone marrow chimeras**. WT or GPR182−/− mice were irradiated with 1100 Rads (2 × 550 Rads) using a gamma irradiator with Cesium-137 as the source before reconstitution with 2 × 10$^6$ total bone marrow cells from WT or GPR182−/− donors[72]. Bone marrow was isolated from the lower extremity long bones of WT or GPR182−/− mice. Irradiated mice received prophylactic antibiotic feed (Uniprim, Neogen) for 24 h prior to irradiation and 2 weeks after irradiation. At 12 weeks post reconstitution, mice were challenged with B16 tumors as described above.

**Tumor chemokine quantification**. Dissected tumor samples were weighed and homogenized in 1 ml 1xPBS containing protease inhibitor using a Fisherbrand$^{TM}$ 150 Handheld homogenizer. The homogenates were centrifuged at 14,000 × *g* for 15 min to collect supernatants. Chemokines in supernatants were subsequently quantified using LEGENDplex Mouse Proinflammatory Chemokine Panel

(BioLegend) and analyzed according to the manufacturer instructions using a Beckman Coulter CytoFlex S. Total protein concentration in each sample was quantified using the Bradford Assay with Coomassie Blue (Bio-Rad) and used to normalize chemokine concentrations to total protein concentration in the supernatant.

**In vitro T cell proliferation**. For examining in vitro T cell proliferation, single-cell suspensions from tumor-draining or contralateral inguinal lymph nodes were harvested. Single-cell suspensions were labeled with CFSE (BioLegend) at a concentration of 1:5000 in PBS for 10 min at room temperature before copious washing with 1× PBS. Cells were plated in flat bottom 96- well plates in RPMI-1640 supplemented with 10% fetal bovine serum at a concentration of 350,000 cells per well. When specified, plates were coated with anti-CD3 (145-2C11; BioLegend) at various concentrations for 24 h and washed twice before cells were seeded. After 96 h of culture, cells were harvested and stained for the following cell surface markers: CD3 (17A2), CD8a (53-6.7), CD4 (RM4-5), and Ghost Dye Red 780 (13-0865-T100, Tonbo Biosciences) before analyses of CFSE dilution using a Beckman Coulter CytoFlex S.

**FITC dextran lymphatic flow assay**. FITC dextran (70 kDa, Sigma-Aldrich) was dissolved in water to a final concentration of 25 mg/mL. 20 μL was injected into the left posterior footpad of WT or GPR182−/− mice. After 30 min mice were euthanized, and the popliteal and inguinal draining lymph nodes were collected. Lymph nodes were disrupted with two 18-gauge needles in 200 μL of PBS to extract soluble FITC dextran for quantification using a Synergy HTX plate reader (BioTek).

**Chemokine binding and internalization**. Conditioned media containing FLAG-human CXCL9 or FLAG-mouse CXCL9 was produced by the Protein Production Shared Resource at the University of Colorado Cancer Center (Aurora, CO). Recombinant human CXCL9 (Biolegend) was labeled with Alexa Flour 488 Microscale Protein Labeling Kit (Thermo Fisher). For flow cytometry binding studies, WT CHO or GPR182 + CHO cells were incubated with 100 μL of conditioned media at 4 °C for 30 min, washed with flow cytometry buffer containing 2% fetal bovine serum, followed by staining with APC anti-DYKDDDDK Tag Antibody (BioLegend) and using a Beckman Coulter CytoFlex S. To assess the GPR182 binding of human chemokines, unlabeled human chemokines at 20 μg/ml were added to compete the binding between GPR182 + CHO cell and Alexa Fluor 488 (AF488)- labeled CXCL9. For CXCL9 $K_d$ calculation, unlabeled CXCL9 was titrated and incubated with GPR182 + CHO cells at 4 °C for 15 min. AF488-labeled CXCL9 was then added at a concentration of 1.3E-6 M, and incubation was continued at 4 °C for an additional 30 min. Cells were then washed 4 times and resuspended for flow cytometric analysis on Cytoflex. GMFI was calculated with FlowJo 10 and entered into Prism 8.0 for non-linear regression analysis and calculation of IC$_{50}$. $K_d$ was calculated by manipulation of the Cheng-Prusoff equation $K_d = IC_{50}/(1 + [competitor]/K_d$ competitor)[73,74] under the assumption that labeled and unlabeled CXCL9 have the same $K_d$. For chemokine internalization, CXCL9-AF488, at 1 μg/ml, was incubated with WT CHO or GPR182 + CHO cells at 37 °C for 20 min, following by paraformaldehyde fixation and co-staining with wheat-germ agglutinin (WGA, Thermo Fischer). For endocytosis studies, unlabeled recombinant human CXCL9, at 1 μg/ml, was incubated with WT 293T or GPR182 + 293T cells at 37 °C for 20 min, following by fixation, permeabilization, and co-staining with Alexa Fluor 647-conjugated anti-GPR182 (ADMR, clone 528563; R&D Systems), primary antibodies against Caveolin-1 (D46G3), anti-Clathrin HC (D3C6), anti-EEA1 (C45B10), or anti-LAMP1 (D2D11) (Cell Signaling) and secondary goat anti-mouse IgG (H + L) secondary antibody (Cy3, Invitrogen). Images were obtained using an Olympus FV100 FCS confocal laser scanning microscope (University of Colorado Anschutz Medical Campus, Advanced Light Microscopy Core Facility, Aurora, CO).

**Immunofluorescence staining, imaging, and analysis**. De-identified primary human melanoma tissue sections (formalin fixed, paraffin embedded) were provided by Dr. William Robinson (University of Colorado, Aurora, CO). All patients provided written informed consent for sample collection according to the Colorado Multiple Institutional Review Board (COMIRB) protocol #05-0309. Human tissue was rehydrated, blocked with 2.5% goat serum, stained with anti-human GPR182 (1:800) (Clone PA5-110928; Invitrogen) overnight at 4 °C followed by goat anti-rabbit IgG (H + L) secondary antibody (Cy5, Invitrogen) for 1 h at room temperature. Following washing, slides were stained with anti-human CD31 (JC/70A, Thermo Fischer)) anti-human PDPN (D2-40, Agilent), or anti-human Prox-1(#ab76696, Abcam) for 1 h at room temperature followed by goat anti-mouse IgG (H + L) secondary antibody (Cy3, Invitrogen). Slides were counterstained with DAPI for 10 min. For multiplex tissue staining, human melanoma tissues were stained for CD31, PDPN, S-100 (Leica Biosystems), and GPR182. Images were taken and analyzed using Vectra Automated Quantitative Pathology Systems at the Human Immunology and Immunotherapy Initiative (HI3).

For mouse tissue samples, tumors were collected and frozen on dry ice in optimum cutting temperature (OCT; Fisher HealthCare) embedding media. The frozen blocks were sectioned at 7 μm and mounted on glass slides. The slides were fixed in acetone, blocked with 2.5% goat serum, and incubated with primary antibodies (anti-mouse CD31 (390), anti-mouse CD3e (145-2C11); BioLegend) for 1 h at room temperature and counterstained with DAPI for 10 min. The slides were then cleared and mounted. Images were taken using a Nikon Eclipse TE2000-E upright microscope and analyzed using SlideBook software (Version 6, Intelligent Imaging Inc). Formalin Fixed Paraffin Embedded tissues were used for mouse GPR182 staining. Sections were rehydrated and boiled in Tris pH9.0 buffer for antigen retrieval. After blocking with 2.5% normal horse serum, the slides were then stained overnight at 4 °C with the following primary antibodies: anti-mouse-GPR182 (1:500, A14854, ABclonal) and anti-mouse-LYVE1–β-gal (1:500, AF2125-SP, R&D). Then, the slides were washed and incubated with secondary antibodies for 1 h at room temperature and counterstained with DAPI for 10 min.

To examine GPR182 protein in human primary LEC cells, human dermal lymphatic endothelial Cells (HDLEC) purchased from Promocell were cultured in Endothelial Cell Growth Medium MV (C-22020, PromoCell) before being fixed, permeabilized, and stained with AF647-conjugated GPR182 mAb (FAB10293R-100, R&D). In some experiments, 500 μM DMOG (Sigma-Aldrich) was added to mimic hypoxic condition. For endocytosis study, HDLECs were incubated with GPR182 mAb-AF647 at 1 μg/ml, 37 °C for 1 h, followed by WGA and DAPI staining.

**Calcium flux and Western blot**. For calcium flux experiments, Fluo-4 Calcium imaging kit was purchased from Invitrogen. HEK293T cells stably expressing GPR182 or CXCR3 and wild-type cells were incubated with Fluo-4 calcium substrate at 37 °C for 30 min, then at room temperature for 30 min, and then washed before flow cytometry analysis using FACSCalibur system and CellQuest Pro (BD). Baseline calcium flux levels were established for each cell line. Recombinant human CXCL11 (BioLegend) was then added to each sample at a concentration of 100 nM, and calcium flux levels were measured for 5 min following ligand incubation.

For Western Blot experiments, HEK293T cells stably expressing GPR182 or CXCR3 and wild type cells were grown in 24-well plates to 60–75% confluence, and then starved with serum-free media for 4 h. Recombinant human CXCL11 (BioLegend) was added to each well at a concentration of 100 nM. After 3 min incubation, cells were washed with PBS and lysed with M-PER Mammalian Protein Extraction Reagent (ThermoFisher) containing PhosSTOP phosphatase inhibitors (Roche). Sample protein concentrations were normalized, combined with Laemmli buffer and 2-Mercaptoethanol (Bio-Rad), and heated at 95 °C for 15 min. Samples were loaded into 12% TGX polyacrylamide gels (Bio-Rad) and proteins were separated via gel electrophoresis and then transferred onto a membrane using Trans-Blot Turbo transfer system (Bio-Rad). The membranes were blocked with 5% milk for 30 min, and then incubated with primary antibodies (phospho-p42/44 MAPK Erk1/2 (D13.14.4E), p44/42 MAPK Erk1/2 (137F5); Cell Signaling) in 4 °C overnight followed by HRP-linked secondary antibodies (Anti-Rabbit HRP-linked; Cell Signaling) for 30 min at room temperature. Membranes were then incubated with Pierce ECL Western Blotting Substrate (ThermoFisher) and imaged using a G:Box Chemi-XX6 camera and GeneSys software (Syngene). In between each primary antibody incubation, membranes were stripped using Restore Western Blot Stripping Buffer (ThermoFisher) for 30 min.

**β-Arrestin recruitment using NanoBit technology**. The NanoBit starter kit was purchased from Promega. We made several constructs to fuse the SmBit of Nluc to the C-end of GPR182 and CXCR3 while the LgBit domain was put at the N-terminal of β-arrestin2. A mixture of constructs was co-transfected into HEK293T cells by Lipofectamine 3000 (Invitrogen). On the second day, 50,000 cells per well were plated into Nunclon Delta Surface 96-well plate (#136101). NanoGLO substrate (from NanoBit kit) was added according to protocol before ligand CXCL11 was then added to each well at a final concentration of 100 nM. Luminescence measurements were recorded immediately and once every minute for total 30 min (BioTek Synergy HTX multimode reader). Data was analyzed in Prism 8.0.

**Single-cell data analysis**. Human melanoma single-cell data was queried using a web-based platform (tisch.comp-genomics.org). The processed data of GSE115978 and GSE72056 were retrieved from GEO website. The ECs with PDPN$^{high}$ and LYVE1$^{high}$ were identified as LEC. The blood ECs were PDPN$^{low}$ and LYVE1$^{low}$ ECs. The expression (TPM) of GPR182 and other ACKRs were compared using the violin plots which were generated by GraphPad Prism 8.0 software.

**Data and statistical analysis**. GraphPad Prism 7.0 and 8.0 software (GraphPad Software) and STATA 15 (StataCorp) were used for all statistical analysis and to generate figures. The Student's t-test and Mann–Whitney tests were performed to compare differences between parametric and non-parametric variables, respectively. When appropriate the one-way analysis of variance (ANOVA) test was performed, prior to individual Student's t-tests with Bonferroni correction for multiple comparisons. The log-rank test was performed to compare overall survival and the repeated measures ANOVA was performed to compare time-dependent tumor growth. All P-values <0.05 were considered to be significant.

**Reporting summary**. Further information on research design is available in the Nature Research Reporting Summary linked to this article.

## Data availability

We used public databases to evaluate GPR182 transcript in human melanoma. The scRNAseq datasets presented in Fig. 1a, b were derived from GSE115978[25] and GSE72056[26]. Data for GPR182 expression and lymphatic score in metastatic melanoma (Supplementary Fig. 1d) was extracted from the TCGA database (https://portal.gdc.cancer.gov). Other relevant data supporting the key findings of this study are available within the article and its Supplementary Information files. The raw numbers for charts and graphs are available in the Source Data file whenever possible. Source data are provided with this paper.

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

## Acknowledgements

We thank Drs William Robinson and Mayumi Fujita at the University of Colorado Anschutz Medical Campus for melanoma tissues and cell lines used in this study. We also thank the University of Colorado Cancer Center, the Human Immune Monitoring Shared Resource, the Advanced Light Microscopy Core Facility, and the Protein Production Shared Resource at the University of Colorado Anschutz Medical Campus for resources used in this study. R.J.T. is supported by NIH/NCATS Colorado CTSI Grant Number UL1 TR002535. Contents are the authors' sole responsibility and do not necessarily represent official NIH views. Y.Z. is supported by the Research Scholar Grant, RSG-17-106-01 LIB, from the American Cancer Society.

## Author contributions

R.J.T. and Y.Z. conceived and designed the project and wrote the manuscript; R.J.T., Y.S., R.L., A.C., Y.F., F.H., E.N.M., R.M.K., and Y.Z. performed the experiments; M.D.N., T.R.L., R.M.K., and R.D.S. analyzed the data and critically reviewed the manuscript. R.D.S. and Y.Z. supervised the project.

## Competing interests

R.J.T., R.D.S., R.M.K., and Y.Z. filed a provisional patent related to this study. The other authors have no competing interests.
