## [Peer Review File · Nature Communications]

GPR182 limits antitumor immunity via chemokine scavenging in mouse melanoma modelsEditorial Note: This manuscript has been previously reviewed at another journal that is not operating a transparent peer review scheme. This document only contains reviewer comments and rebuttal letters for versions considered at *Nature Communications*.

REVIEWER COMMENTS

Reviewer #1 (Remarks to the Author):

Torphy et al. present data showing that GPR182, an ACKR, is expressed in tumor-associated lymphatic vessels and identify its function as a chemokine scavenger in tumors, contributing to immunosuppression. In addition to characterization of this ACKR chemokine binding, they further show increased efficacy of ICB in mouse melanomas in GPR182 KO mice. Novel and interesting findings are presented, providing insight into the function of this chemokine scavenger and lymphatic endothelium in tumor immunity. Overall, the data is of high quality and the conclusions are grounded in the data. Previous studies on which this work builds are taken into the consideration and referred to.

Major comments:

1. Is function of GPR182 in peritumoral lymphatics sufficient for altering chemokine and immune cell composition within the tumors? Tumor models employed have very few intratumoral lymphatics. It is difficult to reconcile how can the loss of GPR182 on LECs, which are mainly outside of the tumor, result in changes in immune cell composition in tumors? Precise localization of lymphatics that express GPR182 in and around the tumor and vessel quantification within the specific areas would provide some insight into the spatial localization of the lymphatics involved in mediating GPR effects.
2. Data clearly established that GPR182 is expressed by lymphatic in tumors. It would be important to know if additional stromal cell types express it or if LECs are exclusive source.
3. Are there any insights into what drives GPR182 upregulation on tumor lymphatics?
4. Authors indicate that there is no obvious change in immune phenotypes in naïve GPR182^{-/-} mice. To better interpret this finding, it would be important to know if and where is GPR182 expressed constitutively under normal conditions. Is GPR182 induced in LECs in other organs as well under certain conditions?
5. Spleen in GPR182^{-/-} mice is enlarged. What is this attributed to, specific increase in certain cell subsets?
6. What happens to tumor growth after day 16 in GPR182^{-/-} mice?
7. Fig. 5D p values don't seem correctly indicated. Only comparison to GPR182^{-/-} can be significant.

Reviewer #2 (Remarks to the Author):

In this study by Trophy and colleagues the authors set out to investigate the function of the orphan receptor GPCR182 in the context of tumors. They establish that Gpr182 is expressed in LEC in human tumors and is further induced under hypoxic conditions. In Gpr182-deficient mice, the tumor growth of several transplanted tumor lines is significantly restrained. The authors further establish that this effect is based on an altered T cell recruitment in particular via CXCR3. Mechanistically, the authors demonstrate that Gpr182 acts as an ACKR by scavenging various chemokines in particular the ligands for CXCR3 (CXCL9, CXCL10 and CXCL11). Finally, the authors

data further imply that targeting GPR182 in combination with checkpoint immunotherapy may be a future strategy to treat non-responsive, cold tumors.

I'm very enthusiastic about this exceptional study! The experiments are technically well performed and the paper is well written and easy to follow. The results are striking with clear differences and phenotypes. The paper encompasses human data, striking observations, new mechanistic insights and the results are highly relevant for a broad readership. The recently published results by Le Mercier et al. are largely in line with this study, yet don't compromise the novelty of the work by Trophy et al. Investigating the function of Gpr182 in the tumor context is novel, highly relevant and will be the basis for numerous studies in the future. Therefore, I recommend publishing this work as soon as possible.

Additional comments:

It appears to me that this manuscript already went through a revision process and as a consequence the Discussion part is already quite extensive. So, I will just share a few thoughts that the authors may or may not integrate into their discussion. Could the changes seen for CCL2, CCL22 and CXCL1 be a consequence of the altered inflammatory milieu due to changes in Teff cell abundance rather than an effect of Gpr182 as a scavenging molecule for these chemokines? To what extent are changes in intratumoral T cell abundance driven by LEC vs BEC expressed Gpr182? Could Gpr182 also change the intratumoral localization and potential cellular interaction partner of Teff cells (e.g. macrophages vs cDC1). Could Gpr182 play a role in guiding tumor cells to lymphatic vessels and therefore promote metastasis?

Reviewer #3 (Remarks to the Author):

In general the authors have addressed the concerns I had on the original manuscript.

The only issue that was not addressed was the need for a schematic diagram. They have referred to other published papers that are at least 5 years old which may lack key recent updates.

REVIEWER COMMENTS

Reviewer #1 (Remarks to the Author):

Torphy et al. present data showing that GPR182, an ACKR, is expressed in tumor-associated lymphatic vessels and identify its function as a chemokine scavenger in tumors, contributing to immunosuppression. In addition to characterization of this ACKR chemokine binding, they further show increased efficacy of ICB in mouse melanomas in GPR182 KO mice. Novel and interesting findings are presented, providing insight into the function of this chemokine scavenger and lymphatic endothelium in tumor immunity. Overall, the data is of high quality and the conclusions are grounded in the data. Previous studies on which this work builds are taken into the consideration and referred to.

Major comments:

1. Is function of GPR182 in peritumoral lymphatics sufficient for altering chemokine and immune cell composition within the tumors? Tumor models employed have very few intratumoral lymphatics. It is difficult to reconcile how can the loss of GPR182 on LECs, which are mainly outside of the tumor, result in changes in immune cell composition in tumors? Precise localization of lymphatics that express GPR182 in and around the tumor and vessel quantification within the specific areas would provide some insight into the spatial localization of the lymphatics involved in mediating GPR effects.

Thanks for the comment. We re-examined our IF staining for GPR182 in our mouse tumor models. The majorities of GPR182+ cells are LYVE1-positive LECs and locate on the edge of the tumors, as shown below. There are some GPR182+ cells within the tumor, though few, which are not LYVE1+ LECs. As shown in the figure below, the majorities of intratumorally immune cells (CD3+ T cells) and lymphatics often locate peritumorally in the tumors. Our co-staining of human melanoma for CXCL9 and GPR182 supports that (Fig. S5F). This close localization makes it possible that LECs regulate chemokines and immune cell composition within the tumors, because chemokines are secreted to disseminate locally in the TME. Because lymphatics are known to function to drain tissue fluid, it is possible that GPR182 on peritumoral lymphatics regulate chemokine levels within the tumor by scavenging chemokines. The specialized location for GPR182 could be the reason why only a few chemokines are increased in GPR182^{-/-} tumors, though GPR182 is able to interact with all chemokines via the GAG-bind motifs. We have included this figure (Fig. S5G, page 12) and Discussions (page 17) in our revised manuscript.

2. Data clearly established that GPR182 is expressed by lymphatic in tumors. It would be important to know if additional stromal cell types express it or if LECs are exclusive source. **Our IF staining data did support that GPR182 is primarily expressed in LECs within tumors. However, lower levels of GPR182 expression in vascular endothelial cells or other stromal cell populations in the tumor microenvironment cannot be fully excluded. In our experimental models, our sensitivity of detecting GPR182 in the tumor microenvironment is limited by antibody detection. We did see some mouse GPR182 expression in non-LEC cells, though few (up Figure). There are reports showing that GPR182 is broadly expressed in vascular endothelium (PMID: 24257808, PMID: 15802528, PMID: 33875597). Furthermore, LECs in LNs constitutively express GPR182. Therefore, although our data supported a role of LECs within the tumor, the possible role of GPR182 on tumor vasculature and on immune responses in lymph node cannot be excluded. To be cautious with our conclusion from experimental results, we have discussed the possible sources of GPR182 in the ‘Discussion’ section (page 15). In addition, we have included the detailed information about GPR182 expression in the Introduction (page 4).**

3. Are there any insights into what drives GPR182 upregulation on tumor lymphatics? **GPR182 has several hypoxia responsive elements (HREs) in the 5' and 3' regulatory region (Hanze, J, et al. PMID: 11883938). Supporting that, our experiment in Fig. S5D (page 11) revealed that cultured human primary LECs upregulated GPR182 expression in the presence of DMOG, an inducer of HIF-1 alpha, to mimic hypoxic conditions. Therefore, hypoxia can be a driving factor to induce GPR182 on tumor lymphatics.**

4. Authors indicate that there is no obvious change in immune phenotypes in naïve GPR182^{-/-} mice. To better interpret this finding, it would be important to know if and where is GPR182 expressed constitutively under normal conditions. Is GPR182 induced in LECs in other organs as well under certain conditions?

GPR182 expression in normal conditions has been carefully studied and described in a few

publications. GPR182 is found to preferentially expressed in the vascular endothelium (PMID: 24257808, PMID: 15802528). Liver and spleen sinusoidal endothelial cells also express GPR182 (PMID: 29408502). In addition, GPR182 is found to be expressed in intestinal stem cells (PMID: 28094771). A recent study generated a BAC Gpr182-mCherry reporter transgenic mouse to carefully examine GPR182 expression in naïve mice. Besides LECs, GPR182 was broadly expressed in vascular endothelial cells of many tissues/organs (PMID: 33875597). We have summarized and included the information and references in the 'Introduction' (page 4).

5. Spleen in GPR182^{-/-} mice is enlarged. What is this attributed to, specific increase in certain cell subsets?

We enumerated the cell composition of the GPR182^{-/-} spleen, with all lymphocyte subtypes increased (Fig. S4C). Consistently, enlarged spleen in GPR182^{-/-} mice has been reported (PMID: 28094771), presumably due to a negative role of GPR182 in hematopoiesis (PMID: 32832870). We have added the information and cited the references in the revision (page 8).

6. What happens to tumor growth after day 16 in GPR182^{-/-} mice?

In B16 and YUMM1.7 tumor models, all tumors in GPR182^{-/-} mice eventually became necrotic or exceed the size limit, when we had to sacrifice the mice based on our animal protocol (Fig. 8B). Only in YUMMER1.7 tumor model, about 30% of GPR182^{-/-} became tumor-free (Fig. 2F). We only included tumor measurements up to day 16, because many tumors became necrotic and we had to sacrifice the mice. Detailed raw data about individual tumor growth can be found in the Data Source we have provided in the of revision.

7. Fig. 5D p values don't seem correctly indicated. Only comparison to GPR182^{-/-} can be significant.

Thanks for pointing out the error which we have corrected in the revision.

Reviewer #2 (Remarks to the Author):

In this study by Trophy and colleagues the authors set out to investigate the function of the orphan receptor GPCR182 in the context of tumors. They establish that Gpr182 is expressed in LEC in human tumors and is further induced under hypoxic conditions. In Gpr182-deficient mice, the tumor growth of several transplanted tumor lines is significantly restrained. The authors further establish that this effect is based on an altered T cell recruitment in particular via CXCR3. Mechanistically, the authors demonstrate that Gpr182 acts as an ACKR by scavenging various chemokines in particular the ligands for CXCR3 (CXCL9, CXCL10 and CXCL11). Finally, the authors data further imply that targeting GPR182 in combination with checkpoint immunotherapy may be a future strategy to treat non-responsive, cold tumors.

I'm very enthusiastic about this exceptional study! The experiments are technically well performed and the paper is well written and easy to follow. The results are striking with clear

differences and phenotypes. The paper encompasses human data, striking observations, new mechanistic insights and the results are highly relevant for a broad readership. The recently published results by Le Mercier et al. are largely in line with this study, yet don't compromise the novelty of the work by Trophy et al. Investigating the function of Gpr182 in the tumor context is novel, highly relevant and will be the basis for numerous studies in the future. Therefore, I recommend publishing this work as soon as possible.

Additional comments:

It appears to me that this manuscript already went through a revision process and as a consequence the Discussion part is already quite extensive. So, I will just share a few thoughts that the authors may or may not integrate into their discussion. Could the changes seen for CCL2, CCL22 and CXCL1 be a consequence of the altered inflammatory milieu due to changes in Teff cell abundance rather than an effect of Gpr182 as a scavenging molecule for these chemokines? To what extent are changes in intratumoral T cell abundance driven by LEC vs BEC expressed Gpr182? Could Gpr182 also change the intratumoral localization and potential cellular interaction partner of Teff cells (e.g. macrophages vs cDC1). Could Gpr182 play a role in guiding tumor cells to lymphatic vessels and therefore promote metastasis?

We thank the reviewer for his/her enthusiasm with our manuscript. In the 'Discussion', we have revised to include thoughts and comments about the possibility of changed TME indirect impact on chemokines (page 17). In our current mouse models, we are unable to evaluate the effects of GPR182 ablation on tumor metastasis, as our animal protocol prevented us to keep mice till metastasis could develop. Thus, the possible impact of GPR182 on lymphatic metastasis has yet to be investigated.

Reviewer #3 (Remarks to the Author):

In general the authors have addressed the concerns I had on the original manuscript. The only issue that was not addressed was the need for a schematic diagram. They have referred to other published papers that are at least 5 years old which may lack key recent updates.

We have provided a schematic diagram (Fig. 8E, page 14) and updated references accordingly.

REVIEWERS' COMMENTS

Reviewer #1 (Remarks to the Author):

All concerns have been addressed, I have no further comments.

Reviewer #2 (Remarks to the Author):

All my concerns have been addressed - congratulations.

Reviewer #3 (Remarks to the Author):

Thank you the authors have addressed all my comments in full.

REVIEWERS' COMMENTS

Reviewer #1 (Remarks to the Author):

All concerns have been addressed, I have no further comments.
We thank for the reviewer's constructive comments.

Reviewer #2 (Remarks to the Author):

All my concerns have been addressed - congratulations.
We thank for the reviewer's constructive comments.

Reviewer #3 (Remarks to the Author):

Thank you the authors have addressed all my comments in full.

We thank for the reviewer's constructive comments.